# Human DUX4 and mouse Dux interact with STAT1 and broadly inhibit interferon-stimulated gene induction

Amy E Spens[1], Nicholas A Sutliff[1], Sean R Bennett[1], Amy E Campbell[2], Stephen J Tapscott[1,3,4]*

[1]Human Biology Division, Fred Hutchinson Cancer Research Center, Seattle, United States; [2]Department of Biochemistry and Molecular Genetics, University of Colorado Anschutz Medical Campus, Denver, United States; [3]Clinical Research Division, Fred Hutchinson Cancer Research Center, Seattle, United States; [4]Department of Neurology, University of Washington, Seattle, United States

**Abstract** DUX4 activates the first wave of zygotic gene expression in the early embryo. Mis-expression of DUX4 in skeletal muscle causes facioscapulohumeral dystrophy (FSHD), whereas expression in cancers suppresses IFNγ induction of major histocompatibility complex class I (MHC class I) and contributes to immune evasion. We show that the DUX4 protein interacts with STAT1 and broadly suppresses expression of IFNγ-stimulated genes by decreasing bound STAT1 and Pol-II recruitment. Transcriptional suppression of interferon-stimulated genes (ISGs) requires conserved (L) LxxL(L) motifs in the carboxyterminal region of DUX4 and phosphorylation of STAT1 Y701 enhances interaction with DUX4. Consistent with these findings, expression of endogenous DUX4 in FSHD muscle cells and the CIC-DUX4 fusion containing the DUX4 CTD in a sarcoma cell line inhibit IFNγ induction of ISGs. Mouse Dux similarly interacted with STAT1 and suppressed IFNγ induction of ISGs. These findings identify an evolved role of the DUXC family in modulating immune signaling pathways with implications for development, cancers, and FSHD.

*For correspondence: stapscot@fredhutch.org

Competing interest: The authors declare that no competing interests exist.

## Editor's evaluation

In this study, the authors provide convincing data to demonstrate that the transcription factor DUX4 functions as a negative regulator of interferon signaling by inhibiting STAT1, thereby suppressing interferon-stimulated gene induction. These studies are important in revealing a critical mechanistic link between DUX4 expression and the modulation of immune signaling pathways. As DUX4 is emerging as a key molecule in early mammalian development and in diverse pathologies including muscular dystrophy and solid tumors, this study will be of broad interest to the fields of development, cancer, and immunology.

## Introduction

Double homeobox (DUX) genes encode a family of transcription factors that originated in placental mammals, consisting of DUXA, DUXB, and DUXC subfamilies that all have similar paired home-odomains. The DUXC family is characterized by a small conserved region at the carboxy-terminus of the protein that includes two (L)LxxL(L) motifs and surrounding conserved amino acids (*Leidenroth and Hewitt, 2010*). Members of this family, including mouse *Dux* and human *DUX4*, are expressed in a brief burst at early stages of development and regulate an initial wave of zygotic gene activation (*De Iaco et al., 2017*; *Hendrickson et al., 2017*; *Whiddon et al., 2017*). While *DUX4* expression has

also been reported in testes and thymus (*Das and Chadwick, 2016*; *Snider et al., 2010*), it is silenced in most somatic tissues.

Mis-expression of *DUX4* in skeletal muscle is the cause of facioscapulohumeral muscular dystrophy (FSHD) (*Campbell et al., 2018*; *Tawil et al., 2014*), the third most prevalent human muscular dystrophy. DUX4 expression in skeletal muscle activates the early embryonic totipotent program, suppresses the skeletal muscle program, and ultimately results in muscle cell loss. Many of the genes induced by DUX4 in skeletal muscle encode proteins that are normally restricted to immune-privileged tissues (*Geng et al., 2012*) and their expression in skeletal muscle could induce an immune response. In this context, it is interesting that FSHD muscle pathology is characterized by focal immune cell infiltrates. However, our prior studies have also suggested that DUX4 might suppress antigen presentation and aspects of an immune response. Expression of DUX4 in cultured muscle cells blocked lentiviral induction of innate immune response genes such as *IFIH1* (*Geng et al., 2012*). More recently, we reported that expression of DUX4 in primary cancers and engineered cancer cell lines blocks the interferon-gamma (IFNγ)-mediated induction of major histocompatibility complex class I (MHC class I) antigen presentation and promotes resistance to immune checkpoint blockade treatments, such as anti-CTLA-4 and anti-PD-1 therapies (*Chew et al., 2019*). The scope and mechanism(s) of how DUX4 suppresses immune signaling remain unknown.

DUX4 contains one LxxLL and one LLxxL motif at its C-terminal end that are among the most highly conserved regions of DUXC family (*Leidenroth and Hewitt, 2010*). LxxLL motifs are alpha-helical protein-interaction domains that were first identified in nuclear-receptor signaling pathways (*Heery et al., 1997*). Proteins containing LxxLL motifs, such as the Protein Inhibitor of Activated STAT or PIAS family, have been shown to modulate immune signaling of STATs, IRFs, NF-kB, and other transcription factors (*Shuai and Liu, 2005*). PIAS proteins block the function of these transcription factors in four ways: preventing DNA binding, recruiting co-repressors, stimulating SUMOylation, or sequestering them within nuclear or subnuclear structures (*Shuai and Liu, 2005*).

In this study, we show that a transcriptionally inactive C-terminal fragment of DUX4 is sufficient to block IFNγ induction of most interferon-stimulated genes (ISGs), and this requires the (L)LxxL(L) domains. Immunoprecipitation and mass spectrometry identified the IFNγ-signaling effector STAT1 and several other proteins involved in immune signaling as proteins that interact with the DUX4 C-terminal domain (DUX4-CTD). We show that the DUX4-CTD interacts with STAT1 phosphorylated at Y701 and interferes with stable DNA binding, recruitment of Pol-II, and transcriptional activation of ISGs. Consistent with these mechanistic studies, endogenous DUX4 in FSHD muscle cells and the CIC-DUX4 fusion protein expressed in a subset of EWSR1-negative small blue round cell sarcomas suppress IFNγ induction of ISGs. The comparable CTD of mouse Dux containing (L)LxxL(L) motifs similarly interacts with STAT1 and blocks IFNγ stimulation of ISGs. These findings suggest an evolved role of the DUXC family in modulating immune signaling pathways and have implications for the role of DUX4 in development, cancers, and FSHD.

## Results

### DUX4 broadly suppresses ISG induction

Our prior studies showed that DUX4 inhibited ISG induction in response to lentiviral infection and suppressed induction of MHC class I proteins in response to IFNγ (type II interferon) (*Chew et al., 2019*; *Geng et al., 2012*). To determine whether DUX4 broadly inhibited ISG induction by IFNγ, we used the MB135-iDUX4 cell line, a human skeletal muscle cell line with an integrated doxycycline-inducible DUX4 (iDUX4) transgene (*Jagannathan et al., 2016*; see *Figure 1—figure supplement 1* for schematics and sequences of the transgenes used in this study). Doxycycline induction of DUX4 expression in the MB135-iDUX4 cell line has been validated as an accurate cell model of the transcriptional consequences of DUX4 expression in FSHD muscle cells (*Jagannathan et al., 2016*) and in the early embryo (*Hendrickson et al., 2017*; *Whiddon et al., 2017*). Using a stringent eightfold induction cut-off (log2 fold-change >3), RNA-seq showed that IFNγ treatment induced 113 genes, whereas the expression of DUX4 suppressed ISG induction by IFNγ more than fourfold for 76 (67%) of these genes and more than twofold for 102 genes (90%) (*Supplementary file 1*).

Informed by the RNA-seq results, we used RT-qPCR to measure the response of four ISGs that represent different components of the response to immune signaling: the RNA helicase *IFIH1*; the

interferon-stimulated exonuclease *ISG20*; the chemoattractant *CXCL9*; and the MHC II chaperone *CD74*. IFNγ induction of all four genes was robustly blocked by DUX4 expression while a DUX4-target gene *ZSCAN4* was strongly induced, indicating that the ISG suppression did not represent a universal block to gene induction (*Figure 1A*, MB135-iDUX4, and *Figure 1—figure supplement 2A* [for this and subsequent constructs, *Figure 1—figure supplement 2* shows RT-qPCR data from additional independent cell lines together with protein expression and nuclear localization]), whereas doxycycline treatment in the absence of iDUX4 did not suppress induction of the ISG panel (*Figure 1A*, MB135 parental). In contrast to DUX4, a paralog in the DUX family, DUXB, did not suppress induction of the ISG panel by IFNγ (*Figure 1A*, MB135-iDUXB).

To determine whether DUX4 also inhibits ISG induction by other innate immune signaling pathways, we transfected the MB135-iDUX4 cells with three different innate immune stimuli: poly(I:C), a long dsRNA mimic to stimulate IFIH1 (MDA5); RIG-I ligand, a short 5'ppp-dsRNA to stimulate DDX58 (RIG-I); or cGAMP, a signaling component of the cGAS dsDNA sensing pathway. Additionally, we stimulated the cells with interferon-beta (IFNβ, type I interferon), which primarily signals through JAK-STAT pathways via a STAT1-STAT2-IRF9 complex, as opposed to the STAT1 homodimers induced by IFNγ. For all signaling pathways, DUX4 suppressed the induction of a subset of the panel of ISG genes induced by each ligand (*Figure 1B*). One exception, *CXCL9* was induced by IFNβ, poly(I:C), and the RIG-I ligand but not suppressed by DUX4. cGAMP did not induce *CXCL9* or *CD74*, precluding evaluation of the role of DUX4 in regulating these ISGs. These results indicate that DUX4 can modulate the activity of multiple signaling pathways. However, because these pathways converge on common nodes, such as the induction of interferon, additional studies are needed to determine whether DUX4 inhibits unique components in each pathway or a common component responsible for ISG upregulation across pathways. We decided to focus further efforts on identifying the mechanism behind the suppression of IFNγ-mediated transcription as this pathway was most broadly suppressed by DUX4.

## DUX4 transcriptional activity is not necessary for ISG suppression

There are two conserved regions of the DUX4 protein, the N-terminal homeodomains (aa19-78, aa94-153) and an ~50 amino acid region at the end of the C-terminal domain (CTD) that is required for transcriptional activation by DUX4 (aa371-424) (*Choi et al., 2016*; *Geng et al., 2011*; *Leidenroth and Hewitt, 2010*). A mutation in the first homeodomain, F67A, significantly diminishes DUX4 DNA binding and target gene activation (*Wallace et al., 2011*). When expressed in MB135 cells, iDUX4-F67A minimally activated the DUX4 target gene *ZSCAN4*, yet still suppressed ISG induction by IFNγ (*Figure 2A and B* and *Figure 1—figure supplement 2B*). A second construct, iDUX4aa154-424, has the N-terminal homeodomain region replaced by a cassette containing a 3x FLAG tag and two nuclear localization signals (3xFLAG-NLS). The iDUX4aa154-424, hereafter called iDUX4-CTD, was completely transcriptionally silent yet equally suppressed activation of ISGs (*Figure 2A and C* and *Figure 1—figure supplement 2C*). RNA sequencing analysis using the same criteria to characterize ISG suppression by the full-length DUX4 demonstrated that the F67A mutant suppressed 70% of induced genes by more than twofold, or 41% of induced genes by more than fourfold, whereas the iDUX4-CTD showed 90 or 52% suppression, respectively (*Supplementary file 1*). Together, these data indicate that DUX4 transcriptional activity is not necessary to suppress IFNγ-mediated gene induction.

## The CTD is necessary and sufficient to suppress ISGs

The DUX4-CTD contains a pair of (L)LxxL(L) motifs, LLDELL and LLEEL, that are conserved across the DUXC/DUX4 family (*Leidenroth and Hewitt, 2010*). DUX4 transgenes with mutations in the first motif, deletion of the second motif, or both (iDUX4mL1, iDUX4dL2, iDUX4mL1dL2) (see *Figure 1—figure supplement 1B* for sequences of these mutants) failed to activate the DUX4 target *ZSCAN4* (*Figure 2A*). iDUX4ml1dl2 and iDUX4dl2 both lost the ability to suppress the panel of ISGs, whereas iDUX4mL1 showed partial activity, suppressing three of the four ISGs (*Figure 2D* and *Figure 1—figure supplement 2D*), indicating that these (L)LxxL(L) motifs are necessary for both ISG suppression and for transcriptional activation by DUX4.

To test sufficiency, we generated two additional C-terminal fragments of DUX4 (*Figure 2C*). The first, iDUX4-CTDmL1dL2, contains the CTD of iDUX4mL1dL2 with its N-terminal HDs replaced with the 3xFLAG-NLS cassette. Similar to iDUX4mL1dL2, iDUX4-CTDmL1dL2 did not block the panel of ISGs (*Figure 2C* and *Figure 1—figure supplement 2E*). The second construct, iDUX4aa339-424, contains

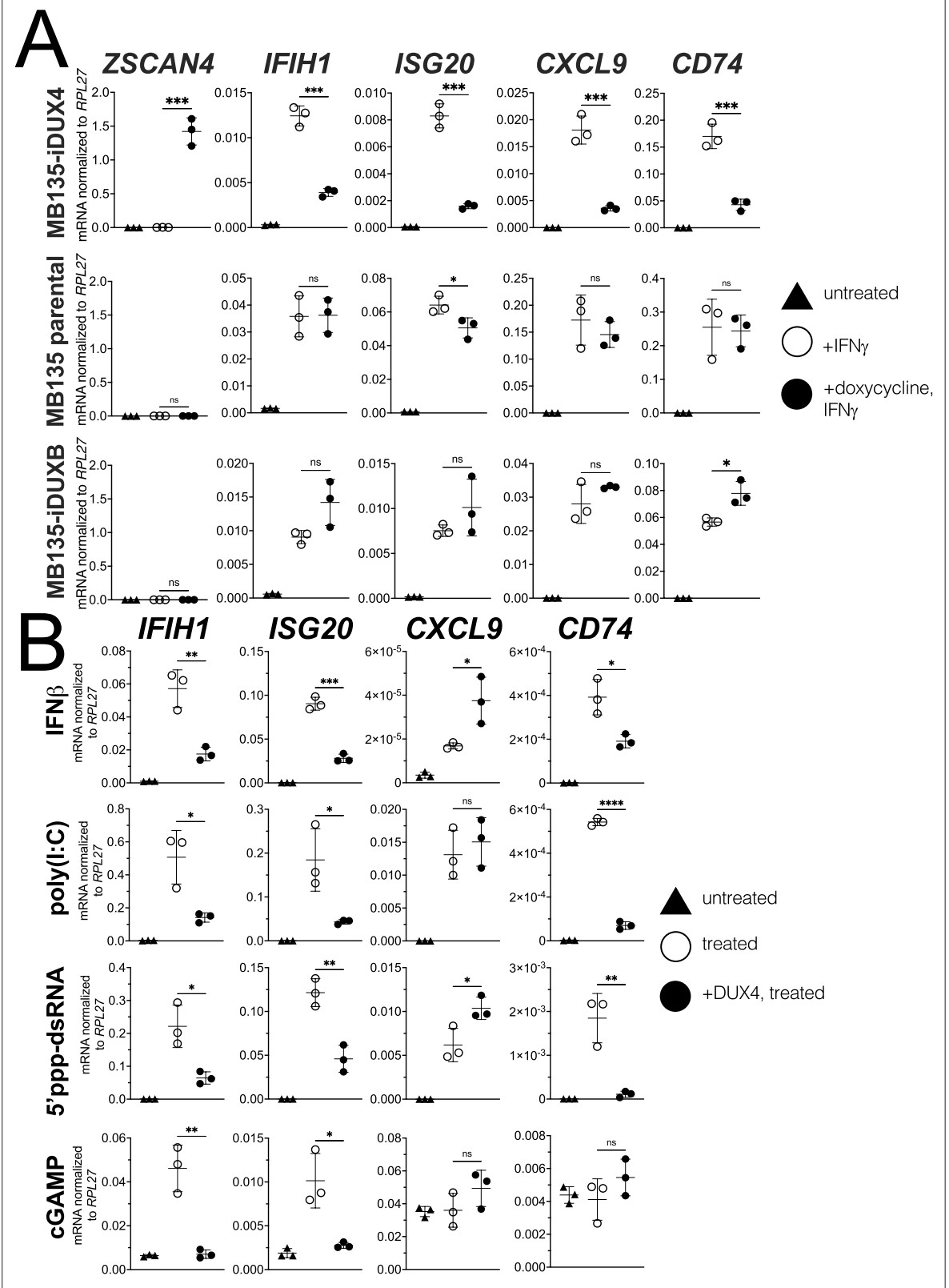

**Figure 1.** DUX4 suppresses interferon-stimulated gene (ISG) induction. (**A**) MB135 cells expressing doxycycline-inducible DUX4 (MB135-iDUX4), parental MB135 cells, or MB135 cells expressing doxycycline-inducible DUXB (MB135-iDUXB) were untreated, treated with IFNγ, or treated with doxycycline and IFNγ. RT-qPCR was used to evaluate expression of a DUX4 target gene, *ZSCAN4*, and ISGs *IFIH1*, *ISG20*, *CXCL9*, and *CD74*. Ct values were normalized to the housekeeping gene *RPL27*. Data represent the mean ± SD of three biological replicates with three technical replicates each. See ***Figure 1—***

*Figure 1 continued on next page*

*Figure 1 continued*

**figure supplement 2** for biological replicates in independent cell lines. (**B**) MB135-iDUX4 cells were untreated, treated with either IFNβ (type 1 IFN pathway), poly(I:C) (IFIH1/MDA5 pathway), 5'ppp-dsRNA (DDX58/RIG-I pathway), or cGAMP (cGAS/STING pathway), or treated with doxycycline and the same immune reagent. RT-qPCR was used to evaluate expression of *IFIH1*, *ISG20*, *CXCL9*, and *CD74*. Ct values were normalized to the housekeeping gene *RPL27*. Data represent the mean ± SD of three biological replicates with three technical replicates each (unpaired *t*-test; ****p<0.0001, ***p<0.001, **p<0.01, *p<0.05, ns p>0.05).

The online version of this article includes the following figure supplement(s) for figure 1:

**Figure supplement 1.** Schematics of constructs cloned for use in this study.

**Figure supplement 2.** Biological replicates in independent cell lines for each DUX4 construct.

only the C-terminal 85 aa residues including both (L)LxxL(L) motifs, and maintained ISG suppression, though not as strongly on the *IFIH1* and *ISG20* genes (*Figure 2C* and *Figure 1—figure supplement 1F*). In summary, these data support a model in which the DUX4-CTD is both necessary and sufficient to suppress a major portion of the ISG response to IFNγ.

## The DUX4 protein interacts with STAT1 and additional immune response regulators

As an unbiased method to identify proteins that interact with the C-terminal region of DUX4, we conducted two experiments using liquid chromatography mass spectroscopy (LC-MS) to identify proteins that co-immunoprecipitated with DUX4-CTD constructs expressed in MB135 myoblasts. We used the DUX4-CTD because the prior experiments showed that it contained the regions necessary and sufficient to suppress ISG induction. In the first experiment, we used MB135iDUX4-CTD cells either untreated, treated with doxycycline alone, or with both doxycycline and IFNγ. In the second experiment, we used MB135iDUX4-CTD and MB135iDUX4mL1dL2 cells both treated with doxycycline and IFNγ compared to these two cell lines untreated and combined as a control. Proteins with a minimum of two peptide spectrum matches (PSMs) in at least one sample that were identified in both experiments were assigned to 1 of 10 categories (see 'Materials and methods') to separate candidate interactors from other categories that might be co-purified because of obligate interactions (e.g., proteasome or ribosome) or might be less likely to be relevant to immune responses (e.g., cytoskeletal proteins). Candidate interactors were then ranked based on the total PSMs for that protein across all samples. (It is important to note that the 'bait' constructs were expressed at low levels in the samples not treated with doxycycline and that the immunoprecipitation concentrated this background, which might account for some of the candidate proteins appearing in the untreated samples.) STAT1 and DDX3X, two key regulators of innate immune signaling, ranked at the top of the list of candidate DUX4-CTD interactors, together with several other proteins implicated in modulating innate immune signaling (*Figure 3*, left panel, and *Supplementary file 2*). Western blot analysis using independent biological samples from a co-IP experiment with MB135-iDUX4-CTD and MB135-iDUXB (as a control) validated the DUX4-CTD interactions with DDX3X, STAT1, PRKDC, YBX1, HNRNPM, PABPC1, NCL, CDK4, and HNRPU (*Figure 3*, right panel).

## The DUX4-CTD preferentially interacts with STAT1 phosphorylated at Y701

Because of its central role in IFNγ signaling, we elected to focus on the interaction of STAT1 with DUX4. To map the region(s) of the DUX4-CTD necessary to interact with STAT1, we expressed a truncation series in MB135 cells (all with an N-terminal 3xFLAG tag and NLS and all treated with IFNγ): iDUX4-CTD (aa154-424), iDUX4aa154-372, iDUX4aa154-308, and iDUX4aa154-271. The region of DUX4 between amino acids 271 and 372 was necessary for co-IP of STAT1, whereas the region between 372 and 424 containing the (L)LxxL(L) motifs might enhance DUX4-CTD binding to the phosphorylated forms of STAT1 (*Figure 4A*).

To determine whether phosphorylation of STAT1 enhanced interaction with DUX4, we co-expressed the FLAG-tagged iDUX4-CTD with an MYC-tagged iSTAT1 or STAT1 mutants Y701A or S727A, wherein doxycycline would induce expression of both the DUX4 and STAT1 transgenes, and performed an αFLAG co-IP to look for STAT1 interaction. The αMYC signal of the IFNγ-treated samples suggests that our iSTAT1, iSTAT1-Y701A, and iSTAT1-S727A transgenes are expressed at similar levels, and

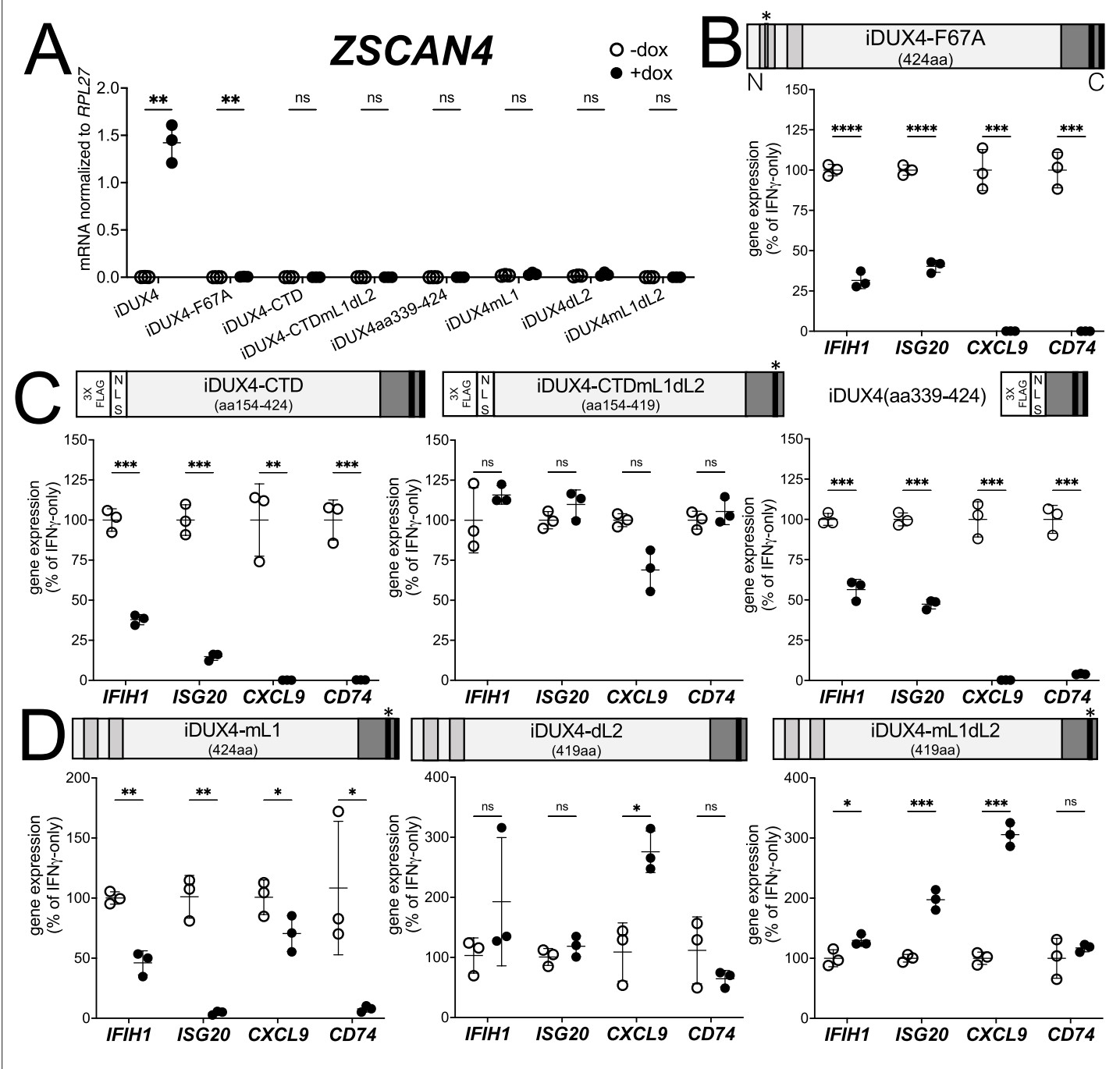

**Figure 2.** DUX4 transcriptional activity is not necessary for interferon-stimulated gene (ISG) suppression, whereas the C-terminal domain (CTD) is both necessary and sufficient. (**A**) MB135 cell lines with the indicated doxycycline-inducible transgene ± doxycycline were evaluated for *ZSCAN4* expression by RT-qPCR as a measure of the ability of the construct to activate a DUX4-target gene. Ct values were normalized to the housekeeping gene *RPL27*. Data represent the mean ± SD of three biological replicates with three technical replicates each. (**B–D**) MB135 cell lines with the indicated doxycycline-inducible transgene were treated with IFNγ ± doxycycline. Light gray, N-terminal boxes, homeodomains; medium gray, C-terminal box, conserved region of CTD; black, C-terminal boxes, (L)LxxL(L) motifs; * indicates sites of mutation for F67A in HD1 and mutation of first LLDELL to AADEAA. See *Figure 1—figure supplement 1* for additional description of 3XFLAG and NLS cassette. RT-qPCR was used to evaluate expression of *IFIH1*, *ISG20*, *CXCL9*, and *CD74* and Ct values were normalized to the housekeeping gene *RPL27*, then normalized to the IFNγ-only treatment to set the induced level to 100%. Data represent the mean ± SD of three biological replicates with three technical replicates each (unpaired *t*-test; ****p<0.0001, ***p<0.001, **p<0.01, *p<0.05, ns p>0.05). See *Figure 1—figure supplement 2* for additional cell lines.

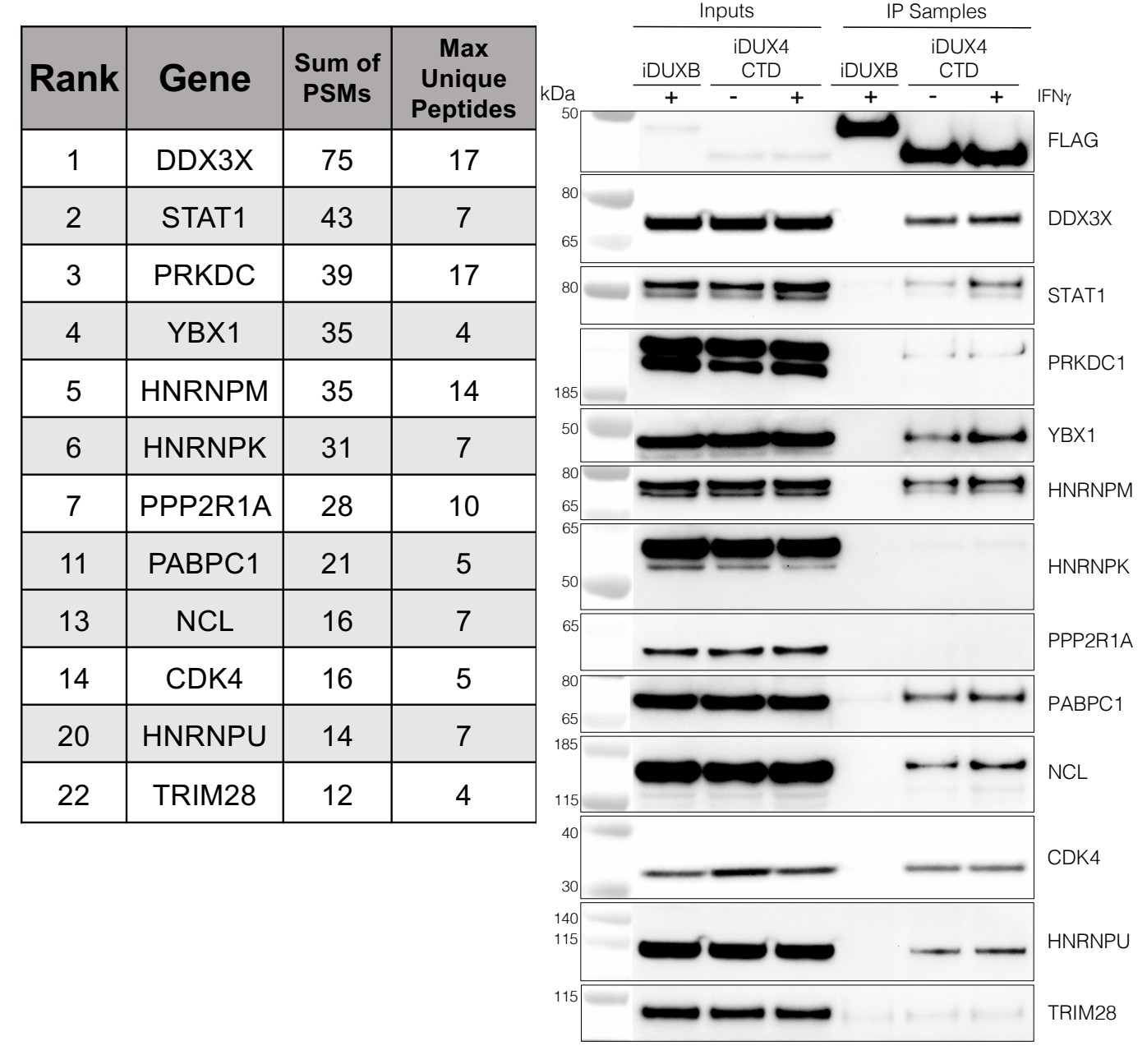

| Rank | Gene | Sum of PSMs | Max Unique Peptides |
|------|------|-------------|---------------------|
| 1 | DDX3X | 75 | 17 |
| 2 | STAT1 | 43 | 7 |
| 3 | PRKDC | 39 | 17 |
| 4 | YBX1 | 35 | 4 |
| 5 | HNRNPM | 35 | 14 |
| 6 | HNRNPK | 31 | 7 |
| 7 | PPP2R1A | 28 | 10 |
| 11 | PABPC1 | 21 | 5 |
| 13 | NCL | 16 | 7 |
| 14 | CDK4 | 16 | 5 |
| 20 | HNRNPU | 14 | 7 |
| 22 | TRIM28 | 12 | 4 |

**Figure 3.** The DUX4 protein interacts with STAT1 and additional immune response regulators. Left panel, representative candidate interactors identified by mass spectrometry of proteins that co-immunoprecipitated with the DUX4-CTD and their relative ranking in the candidate list (see *Supplementary file 2* for full list). Right panel, validation western blot of proteins that co-immunoprecipitate with the DUX4-CTD in cell lysates from MB135 cells expressing doxycycline-inducible 3xFLAG-DUXB or 3xFLAG-DUX4-CTD, ± IFNγ treatment. Data represent biological duplicates. See *Figure 3—source data 1* for uncropped/raw images.

The online version of this article includes the following source data for figure 3:

**Source data 1.** Validation co-IP from inducible MB135 cells lines, anti-FLAG.

**Source data 2.** Validation co-IP from inducible MB135 cell lines, anti-DDX3X.

**Source data 3.** Validation co-IP from inducible MB135 cell lines, anti-STAT1.

**Source data 4.** Validation co-IP from inducible MB135 cell lines, anti-PRKDC.

**Source data 5.** Validation co-IP from inducible MB135 cell lines, anti-YBX1.

**Source data 6.** Validation co-IP from inducible MB135 cell lines, anti-hnRNPM.

*Figure 3 continued on next page*

*Figure 3 continued*

**Source data 7.** Validation co-IP from inducible MB135 cell lines, anti-hnRNPK.

**Source data 8.** Validation co-IP from inducible MB135 cell lines, anti-PPP2R1A.

**Source data 9.** Validation co-IP from inducible MB135 cell lines, anti-PABPC1.

**Source data 10.** Validation co-IP from inducible MB135 cell lines, anti-NCL.

**Source data 11.** Validation co-IP from inducible MB135 cell lines, anti-CDK4.

**Source data 12.** Validation co-IP from inducible MB135 cell lines, anti-hnRNPU.

**Source data 13.** Validation co-IP from inducible MB135 cell lines, anti-TRIM28.

yet the wild-type STAT1 and STAT1-S727A showed enhanced binding to the CTD with IFNγ treatment while the STAT1-Y701A did not (*Figure 4B*). Furthermore, immunofluorescence showed that DUX4-CTD expression, which was highly restricted to the nucleus, did not alter the localization of STAT1 in either untreated cells (low levels throughout the cell) or IFNγ-treated cells (high levels in the nucleus) (*Figure 4—figure supplement 1A and B*). Similarly, there was no detectable pSTAT1-Y701 present in the nuclei of untreated cells, but there was a strong pSTAT1-Y701 signal in IFNγ-treated nuclei (*Figure 4—figure supplement 1C*). Furthermore, the distribution of total STAT1 in the immortalized MB135iDUX4-CTD cells was similar to that in primary human fibroblasts and both immortalized and primary MB135 myoblasts, indicating that the immortalization did not alter the distribution of total STAT1 (*Figure 4—figure supplement 1D*). Proximity ligation assay (PLA) indicated close interaction between the iDUX4-CTD and endogenous pSTAT1-Y701 in the nuclei of MB135 cells treated with doxycycline and IFNγ (*Figure 4C*), and PLA similarly showed an interaction of total STAT1 and DUX4-CTD in primary human fibroblasts (*Figure 4—figure supplement 2*). Therefore, the interaction between DUX4-CTD and STAT1 is enhanced by phosphorylation of STAT1-Y701, and we can observe this interaction within the nuclei of DUX4-CTD-expressing cells.

## The DUX4-CTD decreases STAT1 occupancy at ISG promoters and blocks Pol-II recruitment

Chromatin immunoprecipitation (ChIP) was performed on MB135-iDUX4-CTD cells to assess STAT1 binding to ISG promoters. Compared to a gene-desert region where there should not be STAT1 binding (h16q21), there was a robust induction of STAT1 binding following IFNγ treatment at the promoters of several ISGs (*GBP1*, *IDO1*, *CXCL10*) with previously characterized STAT1 binding sites (*Rosowski et al., 2014*; *Figure 5A*, left four panels). Treatment with IFNγ following induction of DUX4-CTD diminished STAT1 occupancy at all three ISGs, and paired RT-qPCR confirmed that the DUX4-CTD robustly suppressed the RNA induction by IFNγ (*Figure 5A*, right panel). We used CUT&Tag (Cleavage Under Target & Tagmentation) (*Kaya-Okur et al., 2019*) to assess Pol-II occupancy genome wide and found that DUX4-CTD blocked Pol-II recruitment to ISGs without affecting occupancy at other genes (*Figure 5B*).

## Endogenous CIC-DUX4 fusion gene suppresses ISG induction in a sarcoma cell line

The majority of EWSR1 fusion-negative small blue round cell sarcomas have a genetic re-arrangement between CIC and DUX4 that creates a fusion protein containing the carboxyterminal (L)LxxL(L) motif region of DUX4 (*Graham et al., 2012*; *Kawamura-Saito et al., 2006*). We confirmed that the Kitra-SRS sarcoma cell line expresses a CIC-DUX4 fusion mRNA containing the terminal 98 amino acids of DUX4 as previously described (*Nakai et al., 2019*). Compared to MB135 myoblasts, Kitra-SRS cells showed absent-to-low induction of ISGs when treated with IFNγ and control siRNAs. In contrast, siRNA knockdown of the CIC-DUX4 fusion in the KitraSRS cells resulted in a substantially increased IFNγ induction of ISGs, whereas knockdown of CIC in the MB135 cells did not alter ISG induction (*Figure 6A*). To confirm that the CIC-DUX4 fusion was suppressing ISG induction, we expressed a doxycycline-inducible CIC or the Kitra-SRS CIC-DUX4 fusion protein in MB135 cells and showed that the CIC-DUX4 fusion, but not CIC, suppressed IFNγ induction of ISGs *IFIH1*, *CXCL9*, and *CD74*, although not *ISG20* (*Figure 6B*). Furthermore, PLAs were consistent with an interaction of both total

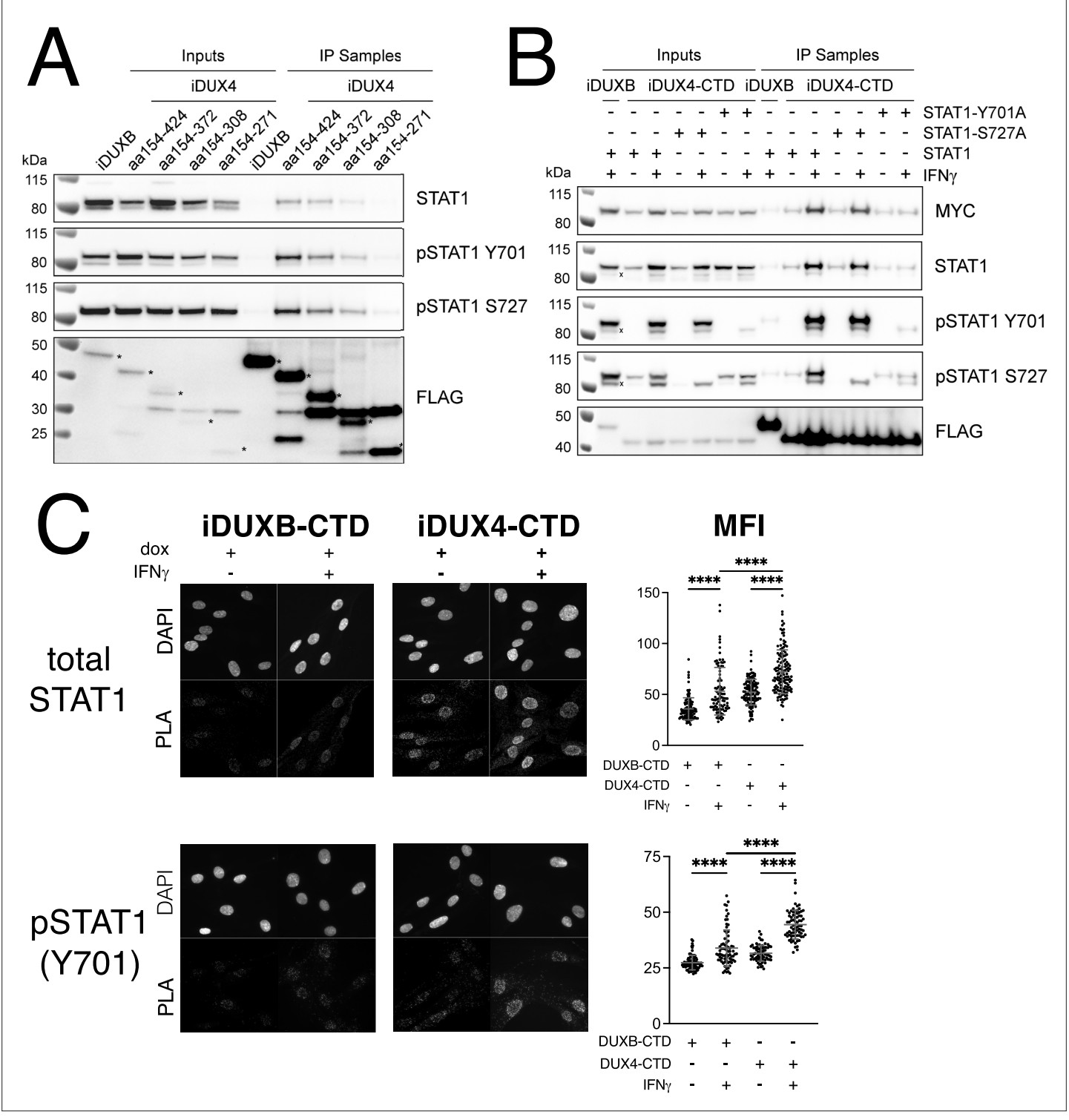

**Figure 4.** The DUX4-CTD preferentially interacts with pSTAT1-Y701. (**A**) Western blot showing input and immunoprecipitated proteins from either 3xFLAG-iDUXB (DUXB) or a truncation series of the 3x-FLAG-iDUX4-CTD cells (iDUX4) precipitated with anti-FLAG and probed with the indicated antibodies. Serial deletions of the iDUX4-CTD were assayed as indicated. All samples were treated with IFNγ.An asterisk indicates the correct band for each FLAG-tagged construct. See **Figure 4—source data 1** for uncropped/raw Western blots. (**B**) Input and anti-FLAG immunoprecipitation from 3xFLAG-iDUXB or 3x-FLAG-iDUX4-CTD cells co-expressing doxycycline-inducible 3xMYC-iSTAT1, -iSTAT1-Y701A, or -iSTAT1-S727A with or without IFNγ treatment and probed with the indicated antibodies. An 'x' indicates the endogenous (non-MYC tagged) STAT1 band. See **Figure 4—source data 1** for uncropped/raw Western blots. (**C**) Proximity ligation assay (PLA) showing co-localization of endogenous STAT1 and pSTAT1 701 with the iDUX4-CTD

*Figure 4 continued on next page*

*Figure 4 continued*

compared to the interaction with the DUXB-CTD, in the nuclear compartment of IFNγ- and doxycycline-treated MB135 cells. Mean fluorescent intensity (MFI) of the nuclei in the PLA channel was measured for 10 images per cell line and treatment and plotted (unpaired *t*-test; ****p<0.0001).

The online version of this article includes the following source data and figure supplement(s) for figure 4:

**Source data 1.** Co-IP from inducible MB135 cell lines, anti-STAT1.

**Source data 2.** Co-IP from inducible MB135 cell lines, anti-pSTAT1(Y701).

**Source data 3.** Co-IP from inducible MB135 cell lines, anti-pSTAT1(S727).

**Source data 4.** Co-IP from inducible MB135 cell lines, anti-FLAG.

**Source data 5.** Co-IP from dual-inducible MB135 cell lines, anti-MYC.

**Source data 6.** Co-IP from dual-inducible MB135 cell lines, anti-STAT1.

**Source data 7.** Co-IP from dual-inducible MB135 cell lines, anti-pSTAT1(Y701).

**Source data 8.** Co-IP from dual-inducible MB135 cell lines, anti-pSTAT1(S727).

**Source data 9.** Co-IP from dual-inducible MB135 cell lines, anti-FLAG.

**Figure supplement 1.** Expression of the DUX4-CTD does not prevent translocation of STAT1 to the nucleus.

**Figure supplement 2.** Primary human foreskin fibroblasts (HFFs) expressing transgenic DUX4CTD show increased interaction with STAT1 and reduced MHC I activation with IFNγ treatment.

---

STAT1 and phosphorylated STAT1-Y701 with the CIC-DUX4 fusion in the nuclei of Kitra-SRS cells (*Figure 6C*).

As DUX4 expression has been shown to dampen MHC I activation in multiple cancer lines (*Chew et al., 2019*), we decided to test the effect of the CIC-DUX4 fusion on MHC I expression in Kitra-SRS cells via flow cytometry. We again treated Kitra-SRS cells and MB135 parental myoblasts with either control siRNAs (siCTRL) or siRNAs targeting CIC and DUX4 (siCIC-DUX4), with or without IFNγ stimulation. We found that knockdown of the endogenous CIC in MB135 myoblasts had no effect on MHC I response to IFNγ (54.9% in siCTRL+ IFNγ compared to 60.7% in siCIC-DUX4+ IFNγ, *Figure 6—figure supplement 1A*). In contrast, knockdown of the CIC-DUX4 fusion protein in Kitra-SRS cells almost doubled the number of MHC I-positive cells with IFNγ treatment (48.1%) compared to cells treated with siCTRL and IFNγ (27.9%, *Figure 6—figure supplement 1B*).

## Endogenous DUX4 expression in FSHD myotubes is associated with suppressed ISGs

DUX4 expression in cultured FSHD muscle cells is often described as low; however, this is due to the high heterogeneity caused by strong expression in a small population of cells (*Rickard et al., 2015*; *Snider et al., 2010*). In cultured FSHD myotubes, approximately 5% of the myotubes might express DUX4 in their nuclei. To determine whether endogenous DUX4 suppresses IFNγ signaling, we assessed induction of IDO1 by IFNγ in FSHD myotubes. Differentiation of FSHD myoblasts into multi-nucleated myotubes results in distinct populations of DUX4-expressing and DUX4-negative myotubes in the same culture, allowing for side-by-side evaluation of DUX4-positive and DUX4-negative muscle cells in the same culture. We determined the IFNγ induction of IDO1 as a representative ISG based on its low basal expression in skeletal muscle and our prior demonstration that it is suppressed in the MB135-iDUX4-CTD cells (see *Figure 5A,* right panel). Treatment with IFNγ produced a reliable IDO1 signal within the nucleus and cytoplasm of individual myotubes that did not express DUX4, whereas DUX4-positive myotubes did not show IDO1 expression in response to IFNγ; we quantified these differences by measuring the mean fluorescent intensity (MFI) of αIDO1 in DUX4+ versus DUX4- nuclei and the MFI of αDUX4 in IDO+ versus IDO1- nuclei and found significant differences by unpaired *t*-test (*Figure 7A*). Therefore, similar to our MB135-iDUX4 studies, endogenous DUX4 expressed at a physiological level is sufficient to prevent ISG induction by IFNγ.

## Conservation of ISG repression and STAT1 interaction in mouse Dux

*Dux*, the mouse ortholog of human *DUX4*, is expressed at the equivalent developmental stage to human *DUX4* (*Hendrickson et al., 2017*), activates a parallel transcriptional program (*Whiddon et al., 2017*), and contains the (L)LxxL(L) motifs that we have shown to be necessary for ISG repression by human DUX4. In fact, the mouse Dux sequence contains a 60 amino acid triplication of the (L)

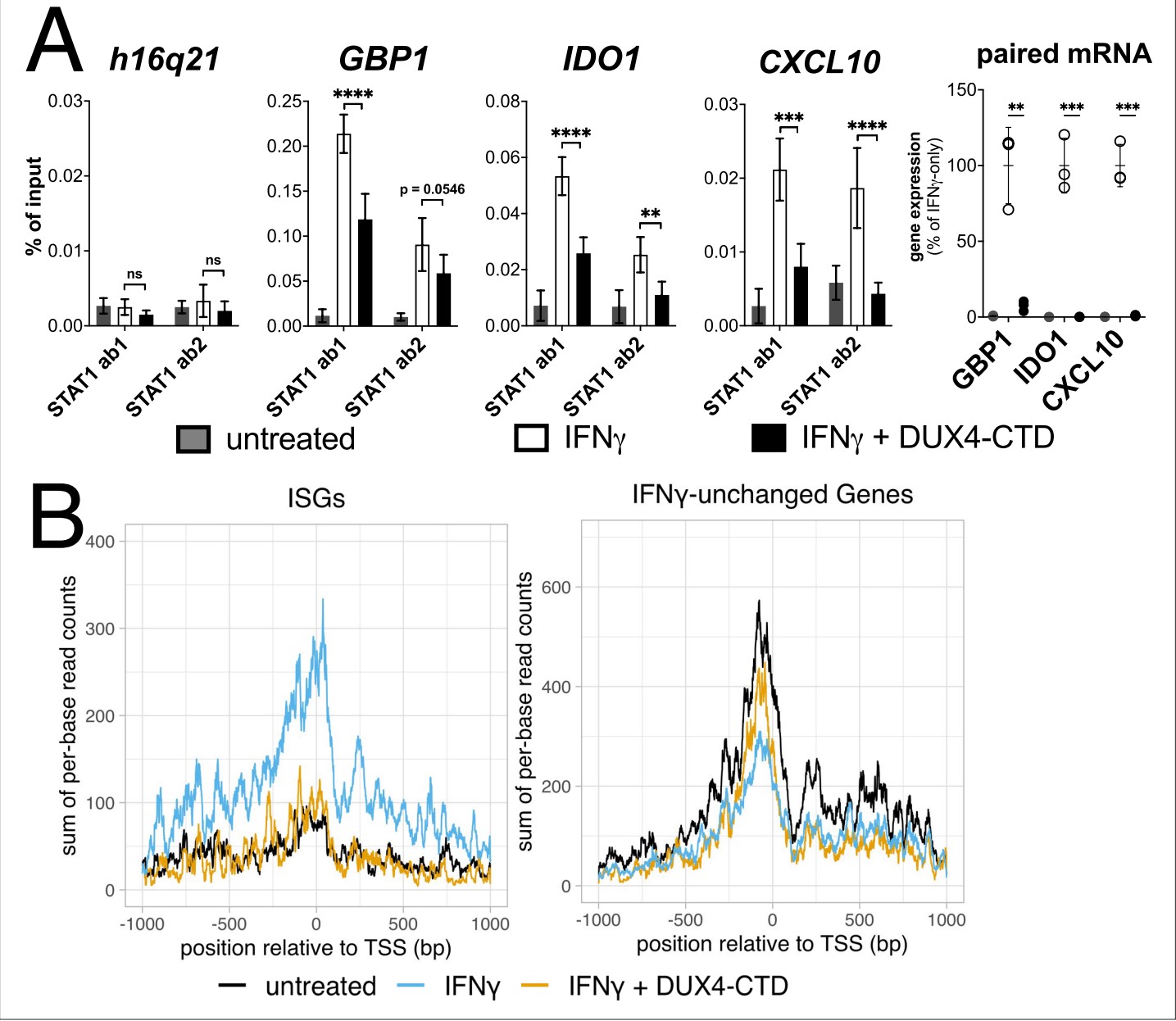

**Figure 5.** The DUX4-CTD decreases STAT1 occupancy at interferon-stimulated gene (ISG) promoters and blocks Pol-II recruitment. (**A**, left four panels) Chromatin immunoprecipitation using anti-STAT1 or IgG from MB135-iDUX4-CTD cells untreated, IFNγ-treated, or IFNγ and doxycycline treated. Ab1: 50:50 mix of STAT1 antibodies Abcam ab239360 and ab234400; Ab2: Abcam ab109320. ChIP-qPCR analysis relative to a standard curve constructed from purified input DNA was used to determine the quantity of DNA per IP sample, which was then graphed as a % of input. Data represent the mean ± SD of two biological replicates with three technical replicates each (unpaired *t*-test; ****p<0.0001, ***p<0.001, **p<0.01, *p<0.05, ns p>0.05). (**A**, right panel) RT-qPCR of RNA from cells used for STAT1 ChIP showing induction of interferon-stimulated genes (ISGs) by IFNγ and suppression by DUX4-CTD. (**B**) CUT&Tag data showing the intensity of Pol-II signal across a 2000 bp window centered on the TSS of ISGs (left) or IFNγ-unchanged genes (right) in untreated, IFNγ-treated, or IFNγ and doxycycline-treated MB135-iDUX4-CTD cells.

LxxL(L)-containing region (*Figure 7—figure supplement 1*). Accordingly, we introduced a doxycycline-inducible mouse *Dux* transgene into human MB135 cells (MB135-iDux) and found that the full-length Dux protein repressed the panel of ISGs even more robustly than the full-length or CTD portion of human DUX4 (*Figure 7B*, left). Similar to human DUX4, Western analysis confirmed the co-immuno-precipitation of STAT1 and both phosphorylated pSTAT1-Y701 and pSTAT1-S727 with mouse Dux (*Figure 7B*, right). These data demonstrate that the suppression of ISG induction and interaction with phosphorylated STAT1 is conserved in the DUXC family.

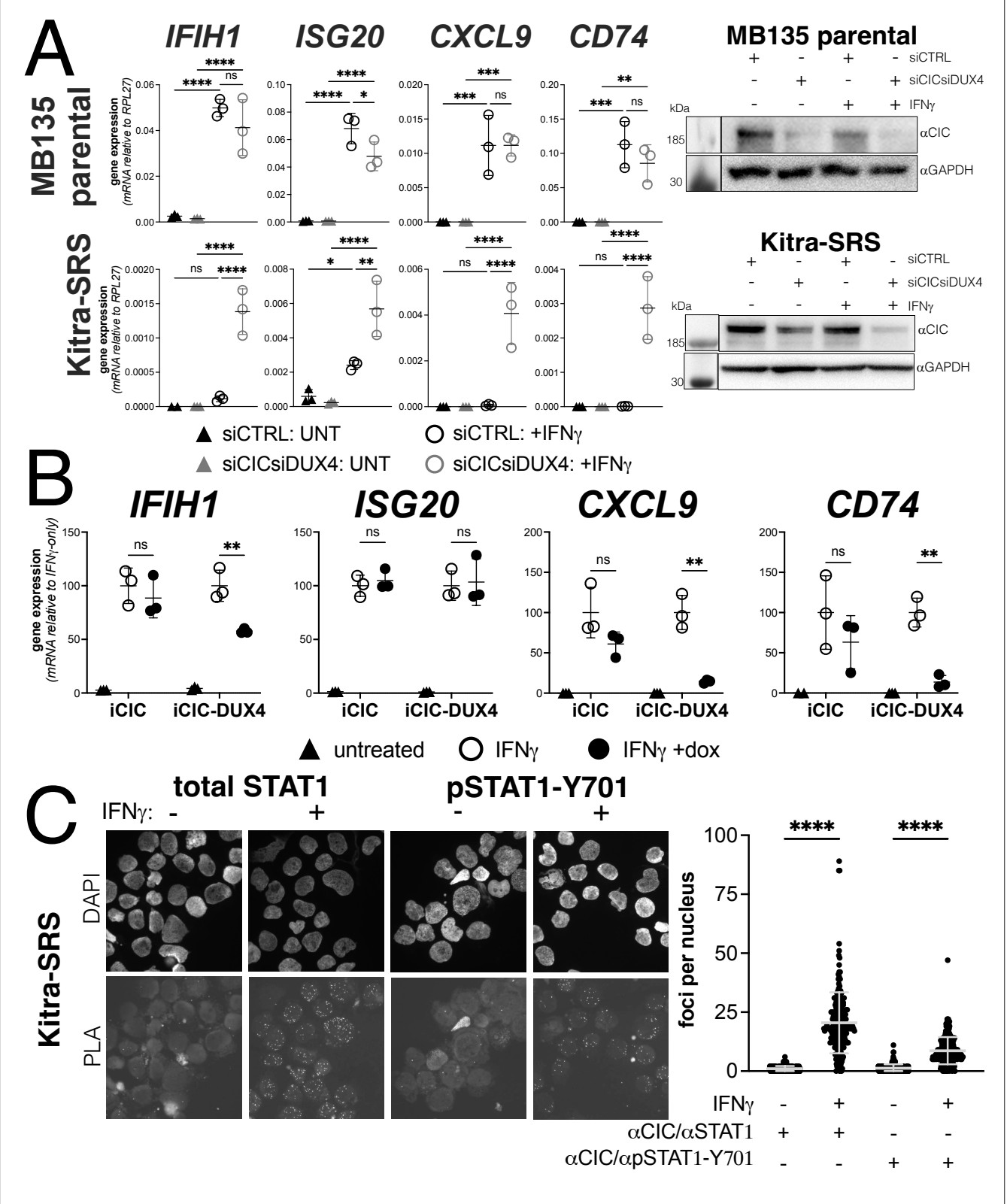

**Figure 6.** Endogenous DUX4 suppresses interferon-stimulated gene (ISG) induction in a sarcoma cell line expressing a CIC-DUX4 fusion gene. (**A**, left panel) RT-qPCR of the indicated genes in MB135 parental or Kitra-SRS that express a CIC DUX4-fusion gene containing the DUX4 CTD. Cells were transfected with control or CIC- and DUX4-targeting siRNAs. Ct values were normalized to the housekeeping gene *RPL27*. Data represent the mean ± SD of three biological replicates with three technical replicates each (unpaired *t*-test; ****p<0.0001, ***p<0.001, ** p<0.01,*p<0.05, ns p>0.05). (**A**, right

*Figure 6 continued on next page*

*Figure 6 continued*

panel) Western blot showing lysates from MB135 or Kitra-SRS cells treated with control or CIC- and DUX4-targeting siRNAs ± IFNγ and probed with the indicated antibodies. See *Figure 6—source data 1* for uncropped/raw western blots. (**B**) RT-qPCR of the indicated genes in MB135 with an inducible CIC (MB135-iCIC) or an inducible CIC-DUX4 fusion gene (MB135-iCIC-DUX4). Cells were untreated, IFNγ-treated, or IFNγ and doxycycline-treated. Ct values were normalized to the housekeeping gene *RPL27*, then normalized to the IFNγ-only treatment to set the induced level to 100%. Data represent the mean ± SD of three biological replicates with three technical replicates each (unpaired *t*-test; \*\*p<0.01, ns p>0.05). (**C**) Proximity ligation assay (PLA) of KitraSRS cells showing association of the endogenous CIC-DUX4 fusion protein with either total STAT1 or phosphorylated STAT1-Y701 exclusively when cells were treated +IFNγ. Mean fluorescent intensity (MFI) was quantified from 200 nuclei per condition and plotted for both pairs of antibodies (unpaired *t*-test; \*\*\*\*p<0.0001).

The online version of this article includes the following source data and figure supplement(s) for figure 6:

**Source data 1.** Parental MB135 anti-CIC.

**Source data 2.** Parental MB135 anti-GAPDH.

**Source data 3.** KitraSRS anti-CIC.

**Source data 4.** KitraSRS anti-GAPDH.

**Figure supplement 1.** Knockdown of the CIC-DUX4 fusion protein in Kitra-SRS cells rescues upregulation of MHC I in response to IFNγ.

## Discussion

In this study, we show that the DUX4-CTD, a transcriptionally inactive carboxyterminal fragment of DUX4, is necessary and sufficient to broadly suppress ISG induction by IFNγ as well as partially inhibit induction through the IFNβ, cGAS, IFIH1/MDA5, and DDX58/RIG-I pathways. The DUX4-CTD colocalizes with STAT1 in the nucleus, diminishes steady-state STAT1 occupancy at ISG promoters, and prevents Pol-II recruitment and transcriptional activation of ISGs by IFNγ. Whereas the conserved DUX4 (L)LxxL(L) motifs are necessary to suppress transcriptional activation by STAT1, they are not necessary for the interaction of DUX4 and STAT1. The suppression of IFNγ signaling by endogenous DUX4 in FSHD muscle cells and the CIC-DUX4 fusion protein in sarcomas provides support for the biological relevance of these findings.

Our data support a simple model of how DUX4 inhibits STAT1 activity (*Figure 7C*). IFNγ binding to its receptor, IFNGR, leads to the phosphorylation of STAT1 at Y701, subsequently STAT1 forms a homodimer, translocates to the nucleus, and binds the gamma-activated sequence (GAS) in the promoters of ISGs. DNA-bound STAT1 is additionally phosphorylated at S727 and recruits Pol-II to the ISG promoters (*Sadzak et al., 2008*; *Wen et al., 1995*). Our studies show that DUX4-CTD interacts with STAT1 phospho-Y701 in the absence of phospho-S727 (i.e., binds the S727A STAT1 mutant), yet also efficiently co-immunoprecipitates with STAT1 phospho-S727 from cell lysates. This indicates that despite DUX4 interacting with STAT1 phospho-Y701, DNA binding of this complex is not fully impaired because of the association with STAT1 phospho-727. However, our ChIP and CUT&Tag studies show decreased STAT1 steady-state occupancy of ISG promoters and failure to recruit Pol-II. Together, these data support a model of DUX4 interaction with pSTAT1-Y701 that prevents the formation of a stable DNA-bound complex and recruitment of Pol-II, but likely not the initial binding of STAT1 to DNA because of the abundance of phospho-S727 associated with DUX4. The (L)LxxL(L) motifs are necessary to prevent transcriptional activation, presumably by blocking Pol-II recruitment, but not necessary for the interaction of DUX4 with STAT1. This could be due to recruitment of a repressor or by simply blocking the interaction of STAT1 with an intermediate factor necessary to recruit Pol-II.

The (L)LxxL(L)-dependent inhibition of STAT1 by DUX4 in this study bears a striking similarity to the inhibitory mechanisms displayed by LxxLL-containing members of the PIAS family. LxxLL motifs were first identified in nuclear-receptor (NR) signaling pathways (*Heery et al., 1997*) where they were found to facilitate protein-protein interactions between unbound NRs and co-repressors such as RIP140 and HDACs, or agonist-bound NRs and co-activators such as CBP/p300 (*Plevin et al., 2005*; *Savkur and Burris, 2004*). LxxLL motifs have since been characterized in multiple protein families, including the PIAS family, and specifically implicated in modulating immune transcriptional networks via interaction with and inhibition of STATs, IRFs, and NF-kB (*Shuai and Liu, 2005*). While the (L)LxxL(L) region of DUX4 is required for suppression of IFNγ-mediated ISG induction and its enhanced interaction with pSTAT1-Y701, it is not required for its apparently weaker interaction with unphosphorylated STAT1. In a similar manner, the LxxLL motif of PIASγ is not required for initial binding to STAT1, but is required to suppress ISG induction mediated by STAT1 in response to both IFNβ (*Kubota et al., 2011*) and

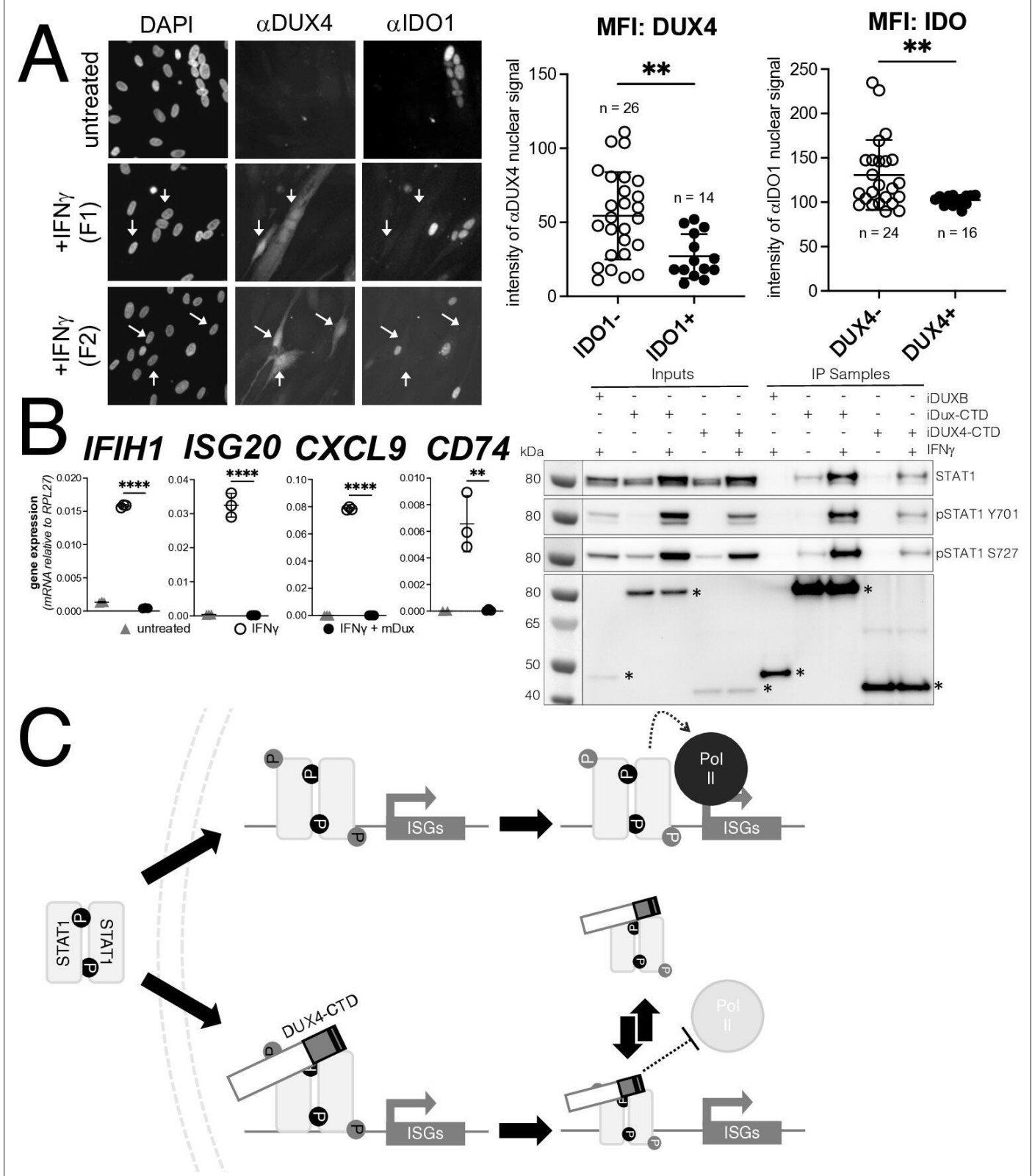

**Figure 7.** Conservation of interferon-stimulated gene (ISG) repression in facioscapulohumeral dystrophy (FSHD) myoblasts and ISG repression and STAT1 interaction by mouse Dux. (**A**) FSHD MB200 myoblasts were differentiated into myotubes, which results in the expression of endogenous DUX4 in a subset of myotubes. Cultures were treated ± IFNγ, and DUX4 and IDO1 were visualized by immunofluorescence. Representative images of untreated and IFNγ-treated (two fields, F1 and F2) cells are shown, with white arrows highlighting DUX4+ myotubes that lack IDO1 signal. Mean fluorescent

*Figure 7 continued on next page*

*Figure 7 continued*

intensity (MFI) of the αDUX4 and αIDO1 nuclear signal was measured in the IFNγ-treated cells only. Data represent the mean ± SD of nuclear MFI from three images, total nuclei per condition listed as 'n' (unpaired *t*-test; **p<0.01). (**B**, left panel) RT-PCR of the indicated genes in MB135-iDux cells untreated or treated with IFNγ ± doxycycline. Ct values were normalized to the housekeeping gene *RPL27*, then normalized to the IFNγ-only treatment to set the induced level to 100%. Data represent the mean ± SD of three biological replicates with three technical replicates each (unpaired *t*-test; ****p<0.0001, **p<0.01). (**B**, right panel) Western blot showing input and immunoprecipitated proteins from either 3xFLAG-iDux or 3x-FLAG-iDUXB cells ± IFNγ precipitated with anti-FLAG and probed with the indicated antibodies. See *Figure 7—source data 1* for uncropped/raw Western blots. (**C**) A model supported by the data showing how the DUX4-CTD might prevent STAT1 ISG induction. (Top) In the absence of the DUX4-CTD, pSTAT1 Y701 (black 'P') dimerizes, translocates to the nucleus, binds its GAS motif in the ISG promoter, acquires secondary phosphorylation at S727 (gray 'P'), and recruits a stable transcription complex that includes Pol-II to drive transcription of ISGs. (Bottom) In the presence of the DUX4-CTD, STAT1 is phosphorylated, translocates to the nucleus, and binds its GAS motif as evidenced by the pSTAT1 S727 in complex with the CTD. However, diminished steady-state occupancy of STAT1 at the ISG promoters and absence of Pol-II recruitment indicate that the STAT1-DUX4-CTD complex does not stably bind DNA and fails to recruit Pol-II and the pre-initiation complex. The (L)LXXL(L) motifs (black bars in DUX4-CTD) are necessary to interfere with transcription suppression and likely prevent STAT1 from interacting with a factor in the pre-initiation complex or recruit a co-repressor.

The online version of this article includes the following source data and figure supplement(s) for figure 7:

**Source data 1.** Mouse Dux co-IP, anti-STAT1.

**Source data 2.** Mouse Dux co-IP, anti-pSTAT1(Y701).

**Source data 3.** Mouse Dux co-IP, anti-pSTAT1(S727).

**Source data 4.** Mouse Dux co-IP, anti-FLAG.

**Figure supplement 1.** Mouse Dux contains a triplication of the (L)LxxL(L)-containing region.

IFNγ (*Liu et al., 2001*). The same motif is required for the trans-repression of androgen receptor (AR) signaling by PIASγ (*Gross et al., 2001*) and of Erythroid Krüppel-like factor (EKLF or KLF1) by PIAS3 (*Siatecka et al., 2015*), though again it is not required for the initial interaction of either pair. The studies referenced above hypothesize that this trans-repression relies on the recruitment of co-repressors, although the specific interactors were not determined. Additionally, just as DUX4 reduces the steady-state occupancy of STAT1 to DNA, PIAS proteins can suppress transcriptional networks by blocking DNA binding, as with PIAS3 and STAT3 (*Chung et al., 1997*) or PIAS1 and NF-kB p65 (*Liu et al., 2005*). These studies describe the mechanisms of transcriptional suppression by LxxLL motifs in PIAS and other proteins that have strong parallels to the (L)LxxL(L) motifs in human DUX4 and mouse Dux. It is important to emphasize that the xx amino acids in the DUXC family are acidic and there is conservation of flanking amino acids as well, suggesting that the DUXC family likely evolved target specificity through these larger areas of conservation.

In addition to STAT1, the mass spectrometry identified several proteins that interact with the DUX4-CTD that might also have a role in modulating immune signaling. Although additional work is needed to validate the biological relevance of these interactions, many have functions related to immune signaling and that will need to be evaluated in future studies. DDX3X and PRKDC are the top-ranked candidates, together with STAT1. DDX3X has been shown to regulate RNA processing, translation, and innate immune signaling (*Mo et al., 2021*). It was also shown to be a pathway-specific regulator of IRF3 and IRF7 in part by acting as a scaffolding factor necessary for IKK-ε and TBK1 phosphorylation of IRFs (*Gu et al., 2013*; *Schröder et al., 2008*). DDX3X was also shown to be a sensor of dsRNA and viral stem-loop RNA with a role in the initial induction of ISGs, including IFIH1 and DDX58 (*Oshiumi et al., 2010*) that then serve to amplify the signaling mechanisms. PRKDC is known mostly for its major roles in DNA repair but also has been implicated in regulating the response to cytoplasmic DNA through the cGAS and IRF3 pathway (*Ferguson et al., 2012*).

Our current findings also provide a molecular mechanism for the suppression of IFNγ-stimulated genes in DUX4-expressing cancers. Previously we reported that the full-length DUX4 is expressed in a diverse set of solid cancers (*Chew et al., 2019*). Cancers expressing DUX4 had diminished IFNγ-induced MHC class I expression, reduced anti-tumor immune cell infiltration, and showed resistance to immune checkpoint blockade. In this study, we show that the CIC-DUX4 fusion in EWSR1-fusion-negative sarcomas blocks IFNγ-induced ISG expression and the upregulation of MHC I. This fusion protein contains the terminal 98 amino acids of DUX4, aa327-424, that encompasses a region shown to be sufficient to suppress IFNγ signaling in the iDUX4-aa339-424 (see *Figure 2C*). It is reasonable to suggest that this fusion protein in the CIC-DUX4 sarcomas, or the full-length DUX4 in some other cancers, contributes to immune evasion at least in part through its interaction with STAT1, and

that targeting DUX4 or its interaction with STAT1 might improve immune-based therapies for DUX4-expressing cancers.

The conservation of the (L)LxxL(L) motifs in mouse Dux and its similar interaction with STAT1 and inhibition of IFNγ signaling indicates that this is a conserved function of the DUXC family. DUX4, Dux, and the canine DUXC all induce expression of endogenous retroelements, as well as pericentromeric satellite repeats that form dsRNAs that, at least in the case of DUX4, induce a dsRNA response that results in activation of PKR and phosphorylation of EIF2α (*Shadle et al., 2019*; *Shadle et al., 2017*). Therefore, it is possible that the interaction with STAT1 and other immune signaling modulators might prevent the activation of the ISG pathway while permitting the PKR response, although the biological consequences remain to be further explored. It is also interesting that DUX4, Dux, and possibly other members of the DUXC family are expressed in immune-privileged tissues – that is, cleavage embryo, testis, and thymus – and our study suggests that their expression might contribute to this immune-privileged state.

It is also important to emphasize the limitations of this study and areas for future research. Although our data show that the DUX4-CTD interacts with STAT1 and prevents Pol-II recruitment to ISGs, further studies will be necessary to determine the mechanism(s). Testing specific steps in the formation of a stable pre-initiation complex might indicate an inhibition of a specific protein interaction necessary for the completion of stable Pol-II recruitment. Although the inhibitory activity of the DUX4-CTD was limited to ISG induction in our experimental system, extending these studies to other signaling pathways and even to artificial gene regulation systems, such as Gal4-Sp1 fusion factors, will be necessary to determine the specificity of the DUX4-CTD activity on STAT1 activity relative to other mechanisms of transcription regulation.

## Materials and methods

### Cell lines

Cell types used: primary human fibroblasts, primary human myoblasts, and immortalized human myoblasts. Myoblasts were obtained from the Fields Center for FSHD Research at the University of Rochester Medical Center. Fibroblasts were obtained from the Fred Hutchinson Cancer Center lab of Dr. D. Miller. Cells were used directly from source. Myoblast identity confirmed by muscle gene expression. Periodic mycoplasma testing did not identify mycoplasma contamination.

### Cell culture

All myoblast experiments were conducted in immortalized MB135 (*Homo sapiens*, female, control, Fields Center for FSHD and Neuromuscular Research at the University of Rochester Medical Center), primary MB135 myoblasts ('MB135 1°,' *H. sapiens*, female, control, Fields Center for FSHD and Neuromuscular Research at the University of Rochester Medical Center), or MB200 (*H. sapiens*, male, FSHD2 subject, Fields Center for FSHD and Neuromuscular Research at the University of Rochester Medical Center) cell lines, respectively, cultured in Ham's F-10 Nutrient Mix (Gibco) supplemented with 15% fetal bovine serum (Hyclone Cat# SH3007103), 100 U/100 µg/ml penicillin/streptomycin (Gibco Cat# 15-140-122), 1 µM dexamethasone (Sigma Cat# D4902), and 10 ng/ml recombinant human basic fibroblast growth factor (PeproTech Cat# G5071). To differentiate the myoblasts to myotubes, media were changed to DMEM supplemented with 10 ug/ml insulin (Sigma Cat# I1882) and 10 ug/ml transferrin (Sigma Cat# T-0665). Cell lines containing doxycycline-inducible transgenes were additionally cultured with 2 µg/ml puromycin (Sigma Cat# P833). Transgenes were induced with 1 µg/ml of doxycycline (Sigma Cat# D9891) for 4 hr prior to other treatments for a total of 20 hr. The Kitra-SRS cells (RRID:CVCL_YI69) were provided by Dr. H. Otani and Osaka University (*Nakai et al., 2019*) and were cultured in DMEM supplemented with 10% fetal bovine serum (Hyclone Cat# SH3007103) and 100 U/100 µg/ml penicillin/streptomycin (Gibco Cat# 15-140-122). Biological replicates consisted of independent but parallel experiments, such as simultaneously stimulating three cell culture plates with IFNγ. Technical replicates consisted of repeat measurements of the same biological sample, such as loading the same biological sample in triplicate for analysis by RT-qPCR.

## Cloning, virus production, and monoclonal cell line isolation

Human DUX4 and mouse Dux truncation constructs were created by cloning synthesized, codon-optimized gBlock fragments into the pCW57.1 vector (Addgene plasmid #41393) downstream of the doxycycline-inducible promoter (replacing the GFP expression gene), or the pRRLSIN vector (Addgene plasmid #12252) downstream of the constitutive hPGK promoter. Lentiviral particles were created by transfecting 293T cells with a subcloned expression vector, the psPAX2 packaging vector (Addgene plasmid #12260), and the pMD2.G envelope vector (Addgene plasmid #12259) using Lipofectamine 2000 according to the manufacturer's instructions (Invitrogen Cat# 11668019). Experimental cell lines were transduced and, when appropriate, selected using 2 μg/ml puromycin at low-enough confluence to allow for isolation of clonal lines using cloning cylinders. Transgenic clonal lines were validated for protein size, expression level, and localization by western blot and immunofluorescence.

## Immune stimulation and RT-qPCR

Myoblasts were transfected with either (final concentrations) 10 μM 2',3'-cGAMP (Invivogen, Cat# tlrl-nacga23), 2 μg/ml poly(I:C) (Sigma, Cat# P1530), or 1 μg/ml 3'ppp-dsRNA RIG-I ligand (Dan Stetson Lab, UW) using Lipofectamine 2000 (Thermo Fisher, Cat# 11668019) according to the manufacturer's protocol or were stimulated with 1000U IFNβ (R&D Systems, Cat# 8499-IF-010-CF) or 200 ng/ml IFNγ (R&D Systems, Cat# 285-IF-100-CF) by addition directly to cell culture medium. After 16 hr of immune stimulation, RNA was collected from cells using the NucleoSpin RNA Kit (Macherey-Nagel, Cat# 740955) according to the manufacturer's instructions. RNA samples were quantified by Nano-Drop and 1 μg of RNA per sample was treated with DNase I Amplification Grade (Thermo Fisher, Cat# 18068015), and then synthesized into cDNA using the Superscript IV First-Strand Synthesis System (Thermo Fisher 18091050), including oligo dT primers (Invitrogen, Cat# 18418012). qPCR was run in 384-well plates on an Applied Biosystems QuantStudio 6 Flex Real-Time PCR System (ABI) and analyzed in Microsoft Excel.

## RNA-seq library preparation and sequencing

RNA was extracted as described above from untreated, doxycycline-treated, IFNγ-treated, or doxycy-cline- and IFNγ-treated samples. RNA was submitted to the Fred Hutchinson Cancer Research Center Genomics Core for library preparation using the TruSeq3 Stranded mRNA kit (Illumina, Cat# RS-122-2001) followed by size and quality analysis by Tapestation (Agilent). Libraries were sequenced on a NextSeq P2-100 (Illumina).

## RNA-seq analysis

Sequencing analysis was performed using R version 4.0.3 (*R Development Core Team, 2020*). Sequencing reads were trimmed using Trimmomatic (version 0.39) (*Bolger et al., 2014*), and aligned to the *H. sapiens* GRCh38 reference genome with the Rsubread aligner (*Liao et al., 2019*). Gene counts were analyzed using featureCounts (v2.0.1) (*Liao et al., 2019*) and the Gencode v35 annotation file. Normalization and differential expression analysis were done with DESeq2 (v1.26.0) (*Love et al., 2014*).

## Immunofluorescence

Cells were fixed for 10 min with 2% paraformaldehyde (Thermo Scientific) for DUX4/STAT1 and 4% paraformaldehyde for DUX4/IDO1, then permeabilized for 10 min with 0.5% Triton X-100 (Sigma), both at room temperature with gentle shaking. Cells were then blocked for 2 hr with PBS/0.3 M glycine/3% BSA at room temperature with gentle shaking. Primary antibodies were incubated at 4°C overnight at the following concentrations: rabbit anti-IDO1 [D5J4E] 1:100 (Cell Signaling Technology, 86630S, RRID:AB2636818), mouse anti-DUX4 [P2G4] 1:250 (*Geng et al., 2011*), rabbit anti-DUX4 [E5-5] 1:1000 (*Geng et al., 2011*), rabbit anti-DUX4 [E14-3] 1:1000 (*Geng et al., 2011*), mouse anti-FLAG [M2] 1:500 (Sigma #F1804, RRID:AB_262044), rabbit anti-STAT1 [EPR4407] 1:750 (Abcam #ab109320, RRID:AB_10863383), and rabbit anti-pSTAT1 Y701 [58D6] 1:400 (Cell Signaling Technology #9167). Cells were washed three times with 1× PBS containing 3% BSA, then secondary antibodies were incubated for 1 hr at room temperature: FITC-conjugated donkey anti-rabbit (Jackson ImmunoResearch #711-095-152, RRID:AB_2315776) or TRITC-conjugated donkey anti-mouse (Jackson

ImmunoResearch #715-025-020, RRID:AB_2340764). Cells were washed once with 1× PBS containing 3% BSA then stained with DAPI (Sigma) 1:5000 for 10 min at room temperature and visualized.

## Fractionated anti-FLAG immunoprecipitation

Cells were lysed on the plate with digitonin lysis buffer pH 7.4 (37.5 µg/ml digitonin, 25 mM Tris-HCl pH 7.5, 125 mM NaCl, 1 mM EDTA, 5% glycerol) supplemented with Pierce Protease Inhibitors EDTA-free (Pierce, Cat# PIA32955) and Pierce Phosphatase Inhibitors (Pierce, Cat# PIA32957), transferred to a centrifuge tube, and incubated for 10 min at 4°C with rotation. Centrifugation at 2500 rcf at 4°C for 5 min pelleted the nuclei, supernatant was discarded, and nuclei resuspended in 1 ml IP buffer pH 7.4 (25 mM Tris-HCl pH 7.5, 175 mM NaCl, 1 mM EDTA, 0.2% NP-40, 5% glycerol) and incubated for 1 hr at 4°C with rotation then spun at 21,000 rcf for 10 min at 4°C to pellet insoluble debris. Protein concentration was determined using the Pierce BCA Protein Assay Kit (Thermo Fisher, Cat# 23225). An equivalent amount of protein per sample was pre-cleared with Dynabeads Protein G beads (Invitrogen, Cat# 10003D) bound to rat anti-mouse IgG for IP (HRP) (Abcam #131368, RRID:AB_2895114) for 1 hr at 4°C with rotation. FLAG-tagged constructs were then immunoprecipitated with Dynabeads Protein G beads coupled to mouse anti-FLAG [M2] (Sigma #F3165, RRID:AB_259529) for 3 hr at 4°C with rotation. Beads were washed 3× with 1 ml IP buffer and eluted by adding 2× NuPage LDS Sample Buffer (Thermo Fisher, diluted from 4× with PBS) to the beads and heating for 10 min at 70°C.

## Liquid chromatography mass spectroscopy (LC-MS)

For LC-MS, anti-FLAG immunoprecipitation was performed with beads cross-linked to mouse anti-FLAG [M2] (Sigma #F3165, RRID:AB_259529) and the proteins competitively eluted with FLAG peptide. Eluted protein samples were electrophoresed into a NuPage 4–12% Bis-Tris gel, excised, and processed by the Fred Hutchinson Cancer Research Center Proteomics Core. Samples were reduced, alkylated, digested with trypsin, desalted, and run on the Orbitrap Eclipse Tribid Mass Spectrometer (Thermo Fisher). Proteomics data were analyzed using Proteome Discoverer 2.4 against a UniProt human database that included common contaminants using Sequest HT and Percolator for scoring. Results were filtered to only include protein identifications from high-confidence peptides with a 1% false discovery rate. Proteins that were identified in at least one sample from both independent experiments with at least two PSMs in one sample were assigned to 1 of 10 categories: 1, candidates; 2, cytoskeletal associated; 3, cytoskeletal; 4, ribosome/translation associated; 5, proteasome associated; 6, membrane or extracellular; ER, golgi, or vesicle associated; 8, lipid metabolism; 9, chaperones; and 10, nuclear import or nuclear membrane associated. The proteins in category 1 were further investigated for interactions with DUX4. It should be noted that this category assignment process de-prioritized groups of proteins based on assignment to a cellular compartment or function (e.g., ribosome/translation proteins might associate with DUX4 as part of a translation complex rather than having a role in immune signaling) and it is possible that some of the proteins assigned to the non-candidate categories might be functional interactors with DUX4 and have an important biological role.

## Chromatin immunoprecipitation and sequencing

ChIP was performed as previously described (*Nelson et al., 2006*) with the following modifications: cells were plated and allowed to grow to 70–80% confluence, then treated with doxycycline and/or IFNγ in combination as labeled in *Figure 5*. Cells were fixed with 1.42% formaldehyde for 15 min at room temperature with shaking. Fixation was quenched with 125 mM glycine, and cells were scraped into Falcon tubes and collected by centrifugation. Cells were lysed to isolate nuclei for 10 min on ice using IP buffer (150 mM NaCl, 50 mM Tris-HCl pH 7.4, 5 mM EDTA, 1% Triton X-100, 0.5% NP-40) containing Pierce Protease Inhibitors EDTA-free (Pierce, Cat# PIA32955) and Pierce Phosphatase Inhibitors (Pierce, Cat# PIA32957) added fresh. Pelleted nuclei were sonicated on a Diagenode Bioruptor on 'Low' for 10 min as 30 s on/30 s off, followed by four rounds of sonication on 'High' for 10 min each as 30 s on/30 s off (50 min total sonication) in IP Buffer + 0.5% SDS. For immunoprecipitation, 500 ng of chromatin was set aside per condition as an 'Input' and 4 µg of antibody was added to 10 µg of chromatin in an equal volume of IP Buffer + 0.5% SDS across samples. 'STAT1 Ab1' consisted of a 50:50 mix of rabbit anti-STAT1 [EPR21057-141] (Abcam #ab234400) and rabbit anti-STAT1 [EPR23049-111] (Abcam ab#239360). 'STAT1 Ab2' was rabbit anti-STAT1 [EPR4407] (Abcam #ab109320, RRID:AB_10863383). For an IgG control, we used purified Rabbit Polyclonal Isotype

Control Antibody (BioLegend #CTL-4112). IP Buffer was added to lower the percentage of SDS <0.1%, and tubes were incubated with rotation overnight at 4°C. During this time, Protein-A Agarose Fast-flow beads (Millipore, Cat# 16-156) were washed twice with IP Buffer and then blocked in IP Buffer containing 2% BSA by rotating overnight at 4°C. After clearing the chromatin as described, beads were aliquoted to fresh tubes and the top 90% of chromatin was transferred to the tubes containing the blocked bead slurry. Tubes were rotated for 1 hr at 4°C. Beads were washed five times with cold IP Buffer containing 0.1% SDS, two times with cold IP Buffer containing 500 mM NaCl, and two times with cold PBS. DNA was isolated as described in the original protocol and used as a template in qPCR. Input DNA was used to create a standard curve. qPCR primers for the h16q21 gene desert region and the ISGs were previously published (*Maston et al., 2012*; *Rosowski et al., 2014*).

## Proximity ligation assay (PLA)

MB135iDUXBCTD, MB135iDUX4CTD, HFF1°, HFF1°-DUXBCTD, and HFF1°-DUX4CTD cells were plated onto Millicell EZ Slide 8-well glass slides (Millipore PEZGS0816) and treated with IFNγ/dox or IFNγ-alone as described in figures. KitraSRS cells were plated onto standard TC dishes and treated ±IFNγ, then trypsinized and scraped from dishes into DMEM to quench the trypsin and pelleted by centrifugation. Pelleted KitraSRS cells were resuspended in 1× PBS and immediately spun onto slides at 1900 rcf for 1 min, after which they were treated identically to the slide-plated cells. All cells were fixed for 10 min with 4% paraformaldehyde (Thermo Scientific), permeabilized for 10 min with 0.5% Triton X-100 (Sigma), and then blocked for 2 hr at room temperature with PBS/0.3 M glycine/3% BSA. Primary antibodies were diluted in PBS/3% BSA and incubated with samples overnight at 4°C. For PLA of cell lines expressing FLAG-tagged transgenes, mouse anti-FLAG [M2] (F1804) (1:4000) was used in combination with either rabbit anti-STAT1 [EPR4407] 1:1000 (Abcam ab109320) or rabbit anti-pSTAT1 Y701 [58D6] 1:1000 (Cell Signaling Technology #9167). For the Kitra-SRS cells expressing endogenous CIC-DUX4, rabbit anti-CIC 1:500 (Invitrogen, PA5-83721) was used in combination with either mouse anti-STAT1 [1/Stat1] 1:1000 (Abcam ab281999) or mouse anti-pSTAT1 Y701 [M135] 1:1000 (Abcam ab29045). Samples were washed three times for 10 min with 1× Wash Buffer A (10 mM Tris, 150 mM NaCl, 0.05% Tween, adjusted pH to 7.4), and then incubated with Duolink In Situ PLA Probe Anti-Rabbit PLUS (Sigma, Cat# DUO92002) and Duolink In Situ PLA Probe Anti-Mouse MINUS (Sigma, Cat# DUO92004) diluted 1:5 in PBS/3% BSA for 1 hr in a humidity chamber at 37°C. Samples were washed three times for 10 min with 1× Wash Buffer A, and then treated with ligase from the Duolink In Situ Detection Reagents Green kit (Sigma, Cat# DUO92014) for 30 min in a humidity chamber at 37°C. Samples were washed three times for 10 min with 1× Wash Buffer A, and then treated with polymerase from the Duolink In Situ Detection Reagents Green kit for 1 hr and 40 min in a humidity chamber at 37°C. Samples were washed two times for 10 min with 1× Wash Buffer B (200 mM Tris, 100 mM NaCl, adjusted pH to 7.5) and then once for 1 min with 0.01× Wash Buffer B. Samples were mounted with Prolong Glass Antifade Mountant with NucBlue (Thermo Fisher, Cat# P36983), and then visualized with a fluorescent microscope using FITC and DAPI filters.

## siRNA knockdown

Cells were transfected with 50 pmol total siRNAs using the Lipofectamine RNAiMax Transfection Reagent (Thermo Fisher, Cat# 13778150) according to the manufacturer's protocol and allowed to sit overnight (16 hr). Cells were changed to fresh growth medium the next morning and allowed to recover during the day, then transfected again with 50 pmol total siRNAs in Lipofectamine RNAiMax Transfection Reagent at the end of the day and left overnight (16 hr). Cells were changed to fresh growth medium the next morning and then used for downstream experiments. The control siRNA (siCTRL) was siOn-Target (Dharmacon, Cat# D-001810-01). The siRNAs targeting CIC-DUX4 were FlexiTube siRNAs Hs_DUX4_11 (QIAGEN, Cat# SI04268453), Hs_CIC_6 (QIAGEN, Cat# SI04275656), and HS_CIC_8 (QIAGEN, Cat# SI04368469).

## CUT&Tag

CUT&Tag was performed as previously described (*Kaya-Okur et al., 2019*) with the following modifications: MB135-iDUX4-CTD myoblasts were plated and allowed to grow to 70–80% confluence. Cells were left untreated, treated with 200 ng/ml IFNγ for 16 hr, or pre-treated with 1 µg/ml doxycycline for 4 hr then had IFNγ added directly to cell media for an additional 16 hr. Fresh cells were harvested

and washed in PBS, crosslinked with 0.1% formaldehyde for 90 s, then counted and 1.25e6 cells were aliquoted per reaction tube. *Drosophila* S2 cells were spiked-in at a genomic ratio of 1:10. Nuclei were prepared from cells in Buffer NE1 (20 mM HEPES-KOH pH 7.9, 10 mM KCl, 0.1% Triton X-100, 20% glycerol, 0.5 mM spermidine, Pierce Protease Inhibitors EDTA-free [PIA32955]) on ice for 10 min and then bound to concanavalin A-coated beads for 10 min. Rabbit anti-phospho Rbp1 CTD (Ser5) [D9N5I] (Cell Signaling Technology, Cat #13523) diluted 1:50 was bound overnight at 4°C in 25 µl per sample of Antibody Buffer (20 mM HEPES-KOH pH 7.5, 150 mM NaCl, 0.5 mM spermidine, 0.01% digitonin, 2 mM EDTA, 1× Roche cOmplete mini EDTA-free protease inhibitor). Anti-rabbit secondary antibody (EpiCypher, Cat#13-0047) diluted 1:100 was bound in 25 µl per sample of Wash150 Buffer (20 mM HEPES-KOH pH 7.5, 150 mM NaCl, 0.5 mM spermidine, 1× Roche cOmplete mini EDTA-free protease inhibitor) for 30 min at room temperature. pAG-Tn5 pre-loaded adapter complexes (EpiCypher, Cat# 15-1017) were added to the nuclei-bound beads for 1 hr at room temperature in 25 µl of Wash300 Buffer (20 mM HEPES-KOH pH 7.5, 300 mM NaCl, 0.5 mM spermidine, 1× Roche cOmplete mini EDTA-free protease inhibitor), then beads were washed and resuspended in Tagmentation Buffer (Wash300 Buffer + 10 mM MgCl$_2$) and incubated at 37°C for 1 hr in a thermocycler with heated lid. Tagmentation was stopped by addition of EDTA, SDS, and proteinase K. DNA was extracted by Phenol-Chloroform and amplified by PCR using CUTANA High Fidelity 2× PCR Master Mix (EpiCypher, Cat#15-1018) and cycling conditions: 5 min at 58°C; 5 min at 72°C; 45 s at 98°C; 14 cycles of 15 s at 98°C, 10 s at 60°C; 1 min at 72°C. PCR products were cleaned up using AMPure XP beads (Beckman Coulter, Cat# A63880) at a ratio of 1.3:1 according to the manufacturer's instructions.

## CUT&Tag analysis

CUT&Tag data were aligned to the GRCh38 patch 13 human genome e following the Benchtop CUT&Tag v3 protocol (*Kaya-Okur et al., 2019*). Subsequent to alignment we calculated 1× genome coverage normalization with read centering and read extension using deepTools' bamCoverage (*Ramírez et al., 2016*) then mapped the resulting coverage tracks to regions of interest using bedtools' map function (*Quinlan and Hall, 2010*). Coverage graphs were plotted using ggplot2 from the tidyverse package in R (*Wickham et al., 2019*).

## Materials availability

Plasmids used in this study will be deposited with Addgene or are available through request to the corresponding author.

## Acknowledgements

We thank Daniel Stetson and Michael Gale for providing advice and reagents, and the FHCRC Genomics and Proteomics Cores for outstanding services. Funding was provided by NIH NIAMS AR045203 (SJT), NCI P30 CA015704 Supplement (SJT), T32 HG000035 (AES), and The Friends of FSH Research and the Chris Carrino Foundation for FSHD (AES, NAS, SRB, AEC, SJT). We thank Dr. H Otani and Osaka University for providing the Kitra-SRS cell line.

## Additional information

### Funding

| Funder | Grant reference number | Author |
|---|---|---|
| National Institute of Arthritis and Musculoskeletal and Skin Diseases | AR045203 | Stephen J Tapscott |
| National Cancer Institute | P30 CA015704 | Stephen J Tapscott |
| National Institutes of Health | T32 HG000035 | Amy E Spens |
| Friends of FSH Research | | Stephen J Tapscott |

| Funder | Grant reference number | Author |
| --- | --- | --- |
| Chris Carrino Foundation | | Stephen J Tapscott |

The funders had no role in study design, data collection and interpretation, or the decision to submit the work for publication.

## Author contributions

Amy E Spens, Conceptualization, Formal analysis, Supervision, Funding acquisition, Validation, Investigation, Visualization, Methodology, Writing - original draft, Project administration, Writing – review and editing; Nicholas A Sutliff, Conceptualization, Formal analysis, Funding acquisition, Validation, Investigation, Visualization, Methodology, Writing - original draft, Writing – review and editing; Sean R Bennett, Conceptualization, Data curation, Validation, Investigation, Visualization, Methodology, Writing - original draft, Writing – review and editing; Amy E Campbell, Conceptualization, Data curation, Validation, Investigation, Methodology; Stephen J Tapscott, Conceptualization, Formal analysis, Supervision, Funding acquisition, Validation, Investigation, Methodology, Project administration, Writing – review and editing

## Author ORCIDs

Amy E Spens http://orcid.org/0000-0002-1902-1309
Amy E Campbell http://orcid.org/0000-0001-6513-5836
Stephen J Tapscott http://orcid.org/0000-0002-0319-0968

## Decision letter and Author response

Decision letter https://doi.org/10.7554/eLife.82057.sa1
Author response https://doi.org/10.7554/eLife.82057.sa2

# Additional files

## Supplementary files

• Supplementary file 1. Processed RNAseq data for MB135iDUX4, MB135iDUX4-F67A, and MB135iDUX4-CTD. Processed RNAseq data for MB135iDUX4, MB135iDUX4-F67A, and MB135iDUX4-CTD myoblasts untreated, treated with IFNγ, or treated with IFNγ following doxycycline-induction of the integrated transgene. Please see 'Materials and methods' for RNAseq analysis description. Raw data have been uploaded to GEO with the identifier GSE186244.

• Supplementary file 2. Processed proteomics data for MB135iDUX4-CTD and MB135iDUX4-CTDmL1dL2. Processed proteomics data for MB135iDUX4-CTD ('longCTD') and MB135iDUX4-CTDmL1dL2 ('mL1dL2') treated and processed as described in 'Materials and methods' under 'Liquid chromatography mass spectroscopy (LC-MS).' Raw data have been deposited to the ProteomeXchange Consortium via the PRIDE partner repository with the dataset identifier PXD029215.

• MDAR checklist

## Data availability

RNA sequencing data and CUT&Tag data are available through GEO GSE186244 and GSE209785, respectively. The mass spectrometry proteomics data have been deposited to the ProteomeXchange Consortium via the PRIDE partner repository with the dataset identifier PXD029215. Standard packages were used for RNA sequencing and CUT&Tag analyses (see 'Materials & Methods'); specific code available via Zenodo.

The following datasets were generated:

| Author(s) | Year | Dataset title | Dataset URL | Database and Identifier |
|---|---|---|---|---|
| Spens AE, Sutliff NA, Bennett SR, Campbell AE, Tapscott SJ | 2022 | RNA sequencing | https://www.ncbi.nlm.nih.gov/geo/query/acc.cgi?acc=GSE186244 | NCBI Gene Expression Omnibus, GSE186244 |
| Spens AE, Sutliff NA, Bennett SR, Campbell AE, Tapscott SJ | 2022 | CUT&Tag | https://www.ncbi.nlm.nih.gov/geo/query/acc.cgi?acc=GSE209785 | NCBI Gene Expression Omnibus, GSE209785 |
| Spens AE, Sutliff NA, Bennett SR, Campbell AE, Tapscott SJ | 2022 | Proteomics | https://www.ebi.ac.uk/pride/archive/projects/PXD029215 | PRIDE, PXD029215 |
| Spens A, Bennett S | 2023 | RNA-seq and Cut&Tag analysis for Human DUX4 and mouse Dux interact with STAT1 and broadly inhibit interferon-stimulated gene induction | https://doi.org/10.5281/zenodo.7927201 | Zenodo, 10.5281/zenodo.7927201 |

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

# Appendix 1

## Appendix 1—key resources table

| Reagent type (species) or resource | Designation | Source or reference | Identifiers | Additional information |
| --- | --- | --- | --- | --- |
| Antibody | Anti-STAT1 (phospho 701) [M135] (mouse monoclonal) | Abcam | Cat# ab29045; RRID:AB_778096 | See 'Materials and methods' for dilution by application |
| Antibody | Anti-STAT1 [1/Stat1] (mouse monoclonal) | Abcam | Cat# ab281999 | See 'Materials and methods' for dilution by application |
| Antibody | Anti-hnRNP M1-M4 [EPR13509(B)] (rabbit monoclonal) | Abcam | Cat# ab177957; RRID:AB_2820246 | See 'Materials and methods' for dilution by application |
| Antibody | Anti-human DNA PKcs [Y393] (rabbit monoclonal) | Abcam | Cat# ab32566; RRID:AB_731981 | See 'Materials and methods' for dilution by application |
| Antibody | Anti-PABPC1 (rabbit polyclonal) | Abcam | Cat# ab21060; RRID:AB_777008 | See 'Materials and methods' for dilution by application |
| Antibody | Anti-STAT1 (phospho S727) [EPR3146] (rabbit monoclonal) | Abcam | Cat# ab109461; RRID:AB_10863745 | See 'Materials and methods' for dilution by application |
| Antibody | Anti-STAT1 [EPR21057-141] (rabbit monoclonal) | Abcam | Cat# ab234400 | See 'Materials and methods' for dilution by application |
| Antibody | Anti-STAT1 [EPR23049-111] (rabbit monoclonal) | Abcam | Cat# ab239360 | See 'Materials and methods' for dilution by application |
| Antibody | Anti-STAT1 [EPR4407] (rabbit monoclonal) | Abcam | Cat# ab109320; RRID:AB_10863383 | See 'Materials and methods' for dilution by application |
| Antibody | Anti-YBX1 [EP2708Y] (rabbit monoclonal) | Abcam | Cat# ab76149; RRID:AB_2219276 | See 'Materials and methods' for dilution by application |
| Antibody | Anti-mouse IgG for IP HRP (rat monoclonal) | Abcam | Cat# AB131368; RRID:AB_2895114 | See 'Materials and methods' for dilution by application |
| Antibody | Isotype control (rabbit polyclonal) | BioLegend | Cat# CTL-4112 | See 'Materials and methods' for dilution by application |
| Antibody | Anti-DDX3X [D19B4] (mouse monoclonal) | Cell Signaling Technology | Cat# 8192; RRID:AB_10860416 | See 'Materials and methods' for dilution by application |
| Antibody | Anti-hnRNP K [R332] (rabbit monoclonal) | Cell Signaling Technology | Cat# 4675; RRID:AB_10622190 | See 'Materials and methods' for dilution by application |
| Antibody | Anti-IDO1 [D5J4E] (rabbit monoclonal) | Cell Signaling Technology | Cat# 86630; RRID:AB_2636818 | See 'Materials and methods' for dilution by application |
| Antibody | Anti-MYC [71D10] (rabbit monoclonal) | Cell Signaling Technology | Cat# 2278; RRID:AB_490778 | See 'Materials and methods' for dilution by application |
| Antibody | Anti-Nucleolin [D4C70] (rabbit monoclonal) | Cell Signaling Technology | Cat# 14574; RRID:AB_2798519 | See 'Materials and methods' for dilution by application |
| Antibody | Anti-phospho Rbp1 CTD (Ser5) [D9N5I] (rabbit monoclonal) | Cell Signaling Technology | Cat# 13523; RRID:AB_2798246 | See 'Materials and methods' for dilution by application |
| Antibody | Anti-PP2A A subunit [81G5] (rabbit monoclonal) | Cell Signaling Technology | Cat# 2041; RRID:AB_2168121 | See 'Materials and methods' for dilution by application |
| Antibody | Anti-pSTAT1 Y701 [58D6] (rabbit monoclonal) | Cell Signaling Technology | Cat #9167 | See 'Materials and methods' for dilution by application |
| Antibody | Anti-TIF1 (TRIM28) [C42G12] (rabbit monoclonal) | Cell Signaling Technology | Cat# 4124; RRID:AB_2209886 | See 'Materials and methods' for dilution by application |
| Antibody | Anti-rabbit secondary antibody (goat mixed monoclonal) | EpiCypher | Cat# 13-0047 | Used in CUT&Tag |
| Antibody | Anti-DUX4 [P2G4] (mouse monoclonal) | *Geng et al., 2011* | N/A | See 'Materials and methods' for dilution by application |

*Appendix 1 Continued on next page*

*Appendix 1 Continued*

| Reagent type (species) or resource | Designation | Source or reference | Identifiers | Additional information |
|---|---|---|---|---|
| Antibody | Anti-DUX4 [E14-3] (rabbit monoclonal) | *Geng et al., 2011* | N/A | See 'Materials and methods' for dilution by application |
| Antibody | Anti-DUX4 [E5-5] (rabbit monoclonal) | *Geng et al., 2011* | N/A | See 'Materials and methods' for dilution by application |
| Antibody | Anti-mouse IgG HRP (goat superclonal) | Invitrogen | Cat# A28177 | See 'Materials and methods' for dilution by application |
| Antibody | Anti-CIC (rabbit polyclonal) | Invitrogen | Cat# PA5-83721 | See 'Materials and methods' for dilution by application |
| Antibody | FITC-conjugated anti-rabbit (donkey monoclonal) | Jackson ImmunoResearch | Cat# 711-095-152; RRID:AB_2315776 | See 'Materials and methods' for dilution by application |
| Antibody | TRITC-conjugated anti-mouse (donkey monoclonal) | Jackson ImmunoResearch | Cat# 715-025-020; RRID:AB_2340764 | See 'Materials and methods' for dilution by application |
| Antibody | Anti-CDK4 (rabbit polyclonal) | ProteinTech | Cat# 11026-1-AP; RRID:AB_2078702 | See 'Materials and methods' for dilution by application |
| Antibody | Anti-HAT1 (rabbit polyclonal) | ProteinTech | Cat# 11432-1-AP; RRID:AB_2116435 | See 'Materials and methods' for dilution by application |
| Antibody | Anti-HNRNPU (rabbit polyclonal) | ProteinTech | Cat# 14599-1-AP; RRID:AB_2248577 | See 'Materials and methods' for dilution by application |
| Antibody | Anti-FLAG [M2] (mouse monoclonal) | Sigma-Aldrich | Cat# F1804; RRID:AB_262044 | See 'Materials and methods' for dilution by application |
| Antibody | Anti-FLAG [M2] (mouse monoclonal) | Sigma-Aldrich | Cat# F3165; RRID:AB_259529 | See 'Materials and methods' for dilution by application |
| Antibody | Anti-rabbit IgG HRP (goat superclonal) | Thermo Fisher | Cat# A27036; RRID:AB2536099 | See 'Materials and methods' for dilution by application |
| Cell line (*Homo sapiens*) | MB200 (male, FSHD2), immortalized | Fields Center for FSHD and Neuromuscular Research | https://www.urmc.rochester.edu/neurology/fields-center.aspx | |
| Cell line (*H. sapiens*) | MB135 (female), immortalized | *Geng et al., 2012* | N/A | |
| Cell line (*H. sapiens*) | MB135-iDUX4 (SSc7, female) | *Jagannathan et al., 2016* | N/A | |
| Cell line (*H. sapiens*) | HFF-DUX4CTD | This study | N/A | Primary HFF cells transduced with the constitutive pRRLSIN-3XFLAG-NLS-DUX4CTD lentiviral expression construct |
| Cell line (*H. sapiens*) | HFF-DUXB-CTD | This study | N/A | Primary HFF cells transduced with the constitutive pRRLSIN-3XFLAG-NLS-DUXBCTD lentiviral expression construct |
| Cell line (*H. sapiens*) | MB135-i3XFLAG-CIC (female) | This study | N/A | Immortalized MB135 myoblasts transduced with the specified inducible lentiviral expression construct |
| Cell line (*H. sapiens*) | MB135-i3XFLAG-CIC-DUX4 (female) | This study | N/A | Immortalized MB135 myoblasts transduced with the specified inducible lentiviral expression construct |
| Cell line (*H. sapiens*) | MB135-iDux-CA (female) | This study | N/A | Immortalized MB135 myoblasts transduced with the specified inducible lentiviral expression construct |
| Cell line (*H. sapiens*) | MB135-iDux-CTD (female) | This study | N/A | Immortalized MB135 myoblasts transduced with the specified inducible lentiviral expression construct |
| Cell line (*H. sapiens*) | MB135-iDUX4 (ASc4, female) | This study | N/A | Immortalized MB135 myoblasts transduced with the specified inducible lentiviral expression construct |
| Cell line (*H. sapiens*) | MB135-iDUX4 (NSc2, female) | This study | N/A | Immortalized MB135 myoblasts transduced with the specified inducible lentiviral expression construct |

*Appendix 1 Continued on next page*

*Appendix 1 Continued*

| Reagent type (species) or resource | Designation | Source or reference | Identifiers | Additional information |
|---|---|---|---|---|
| Cell line (*H. sapiens*) | MB135-iDUX4-CTD (AES150-1, female) | This study | N/A | Immortalized MB135 myoblasts transduced with the specified inducible lentiviral expression construct |
| Cell line (*H. sapiens*) | MB135-iDUX4-CTD (AES150-5, female) | This study | N/A | Immortalized MB135 myoblasts transduced with the specified inducible lentiviral expression construct |
| Cell line (*H. sapiens*) | MB135-iDUX4-CTDmL1dL2 (AES150-1, female) | This study | N/A | Immortalized MB135 myoblasts transduced with the specified inducible lentiviral expression construct |
| Cell line (*H. sapiens*) | MB135-iDUX4-CTDmL1dL2 (AES150-3, female) | This study | N/A | Immortalized MB135 myoblasts transduced with the specified inducible lentiviral expression construct |
| Cell line (*H. sapiens*) | MB135-iDUX4-F67A (ASc10, female) | This study | N/A | Immortalized MB135 myoblasts transduced with the specified inducible lentiviral expression construct |
| Cell line (*H. sapiens*) | MB135-iDUX4-F67A (ASc6, female) | This study | N/A | Immortalized MB135 myoblasts transduced with the specified inducible lentiviral expression construct |
| Cell line (*H. sapiens*) | MB135-iDUX4aa154-271 (female) | This study | N/A | Immortalized MB135 myoblasts transduced with the specified inducible lentiviral expression construct |
| Cell line (*H. sapiens*) | MB135-iDUX4aa154-308 (female) | This study | N/A | Immortalized MB135 myoblasts transduced with the specified inducible lentiviral expression construct |
| Cell line (*H. sapiens*) | MB135-iDUX4aa154-372 (female) | This study | N/A | Immortalized MB135 myoblasts transduced with the specified inducible lentiviral expression construct |
| Cell line (*H. sapiens*) | MB135-iDUX4aa339-424 (NSc10, female) | This study | N/A | Immortalized MB135 myoblasts transduced with the specified inducible lentiviral expression construct |
| Cell line (*H. sapiens*) | MB135-iDUX4aa339-424 (NSc5, female) | This study | N/A | Immortalized MB135 myoblasts transduced with the specified inducible lentiviral expression construct |
| Cell line (*H. sapiens*) | MB135-iDUX4aa339-424 (NSc8, female) | This study | N/A | Immortalized MB135 myoblasts transduced with the specified inducible lentiviral expression construct |
| Cell line (*H. sapiens*) | MB135-iDUX4dL2 (NSc1, female) | This study | N/A | Immortalized MB135 myoblasts transduced with the specified inducible lentiviral expression construct |
| Cell line (*H. sapiens*) | MB135-iDUX4mL1 (NSc3, female) | This study | N/A | Immortalized MB135 myoblasts transduced with the specified inducible lentiviral expression construct |
| Cell line (*H. sapiens*) | MB135-iDUX4mL1dL2 (NSc2, female) | This study | N/A | Immortalized MB135 myoblasts transduced with the specified inducible lentiviral expression construct |
| Cell line (*H. sapiens*) | MB135-iDUX4mL1dL2 (NSc3, female) | This study | N/A | Immortalized MB135 myoblasts transduced with the specified inducible lentiviral expression construct |
| Cell line (*H. sapiens*) | MB135-iDUX4mL1dL2 (NSc8, female) | This study | N/A | Immortalized MB135 myoblasts transduced with the specified inducible lentiviral expression construct |
| Cell line (*H. sapiens*) | MB135-iDUXB (female) | This study | N/A | Immortalized MB135 myoblasts transduced with the specified inducible lentiviral expression construct |
| Cell line (*H. sapiens*) | MB135 (female), primary | Dr. Rabi Tawil, Fields Center for FSHD Research, University of Rochester Medical Center | N/A | Primary myoblast cells derived from patient muscle biopsy sample |
| Cell line (*H. sapiens*) | Primary human foreskin fibroblasts ('HFF,' male) | Dr. Dusty Miller, Fred Hutchinson Cancer Center | N/A | Primary human foreskin fibroblast cells derived from patient foreskin tissue |
| Chemical compound, drug | RIG-I ligand | Gift of Dr. Dan Stetson Lab, UW | N/A | |

*Appendix 1 Continued on next page*

*Appendix 1 Continued*

| Reagent type (species) or resource | Designation | Source or reference | Identifiers | Additional information |
|---|---|---|---|---|
| Chemical compound, drug | 2'3'-cGAMP | Invivogen | Cat# tlrl-nacga23 | |
| Chemical compound, drug | Recombinant human IFN-beta protein | R&D Systems | Cat# 8499-IF-010-CF | |
| Chemical compound, drug | Recombinant human IFN-gamma | R&D Systems | Cat# 285IF100CF | |
| Chemical compound, drug | Polyinosinic-polycytidylic acid sodium salt [poly(I:C)] | Sigma | Cat# P1530 | |
| Commercial assay or kit | iTaq SYBR Green Supermix | Bio-Rad | Cat# 1725124 | |
| Commercial assay or kit | CUTANA Non-Hot Start 2X PCR Master Mix for CUT&Tag | EpiCypher | Cat# 15-1018 | Used in CUT&Tag |
| Commercial assay or kit | CUTANA pAG-Tn5 for CUT&Tag | EpiCypher | Cat# 15-1017 | Used in CUT&Tag |
| Commercial assay or kit | Illumina TruSeq RNA Sample Prep v2 Kit | Illumina | Cat# RS-122-2001 | |
| Commercial assay or kit | Dnase Amp grade | Invitrogen | Cat# 18068015 | |
| Commercial assay or kit | Oligo(dT) 12–18 primer | Invitrogen | Cat# 18418012 | |
| Commercial assay or kit | RNaseOUT Recombinant Ribonuclease Inhibitor | Invitrogen | Cat# 10777019 | |
| Commercial assay or kit | Superscript IV | Invitrogen | Cat# 18091050 | |
| Commercial assay or kit | Lipofectamine RNAiMAX | Life Technologies | Cat# 13778150 | |
| Commercial assay or kit | NucleoSpin RNA kit | Macherey-Nagel | Cat# 740955 | |
| Commercial assay or kit | Lipofectamine 2000 | Thermo Fisher | Cat# 11668019 | |
| Commercial assay or kit | Superscript IV First-Strand Synthesis System | Thermo Fisher | Cat# 18091050 | |
| Other | Agencourt AMPure XP beads | Beckman Coulter | Cat# A63880 | Used in CUT&Tag |
| Other | Hyclone FBS | Fisher | Cat# SH3007103 | Used to supplement F-10 for cell culture of myoblast lines |
| Other | Gibco Penicillin-Streptomycin (10,000 U/ml) | Fisher Scientific | Cat# 15-140-122 | Anti-fungal to supplement cell culture media |
| Other | Dynabeads Protein G beads | Invitrogen | Cat# 10003D | Used in fractionated anti-FLAG immunoprecipitation |
| Other | ProLong Glass antifade Mountant with Nucblue | Invitrogen | Cat# P36983 | Used to mount slides for proximity ligation assays |
| Other | Millicell EZ Slide 8-well glass slides | MilliporeSigma | Cat# PEZGS0816 | Used to culture cells for proximity ligation assays |
| Other | Protein-A agarose beads | MilliporeSigma | Cat# 16-156 | Used in ChIP-qPCR |
| Other | Pierce phosphatase inhibitors | Pierce | Cat# PIA32957 | Used in ChIP-qPCR, CUT&Tag |
| Other | Pierce protease inhibitors (EDTA-free) | Pierce | Cat# PIA32955 | Used in ChIP-qPCR, CUT&Tag |
| Other | Recombinant human basic fibroblast growth factor | Promega | Cat# G5071 | Used to supplement F-10 for cell culture of myoblast lines |

*Appendix 1 Continued on next page*

*Appendix 1 Continued*

| Reagent type (species) or resource | Designation | Source or reference | Identifiers | Additional information |
|---|---|---|---|---|
| Other | Dexamethasone | Sigma-Aldrich | Cat# D4902 | Used to supplement F-10 for cell culture of myoblast lines |
| Other | Doxycycline hyclate | Sigma-Aldrich | Cat# D9891 | Used to induce doxycycline-inducible transgenes |
| Other | Duolink In Situ Detection Reagents Green kit | Sigma-Aldrich | Cat# DUO92014 | Used in PLA |
| Other | Duolink In Situ PLA Probe Anti-Mouse MINUS | Sigma-Aldrich | Cat# DUO92004 | Used in PLA |
| Other | Duolink In Situ PLA Probe Anti-Rabbit PLUS | Sigma-Aldrich | Cat# DUO92002 | Used in PLA |
| Other | Insulin | Sigma-Aldrich | Cat# I1882 | Used in differentiating MB200 myoblasts into myotubes |
| Other | Polybrene | Sigma-Aldrich | Cat# 107689 | Used in transducing cell lines with lentivirus |
| Other | Puromycin dihydrochloride | Sigma-Aldrich | Cat# P833 | Used as a selective agent for puromycin-resistant cell lines |
| Other | Transferrin | Sigma-Aldrich | Cat# T-0665 | Used in differentiating MB200 myoblasts into myotubes |
| Other | OptiMEM Reduced Serum Medium | Thermo Fisher | Cat# 31985070 | Used for lipofection |
| Recombinant DNA reagent | pMD2.G | Didier Trono Lab | Addgene#12259; RRID:Addgene_12259 | VSV-G envelope expressing plasmid |
| Recombinant DNA reagent | psPAX2 | Didier Trono Lab | Addgene#12260; RRID:Addgene_12260 | Lentiviral packaging plasmid |
| Recombinant DNA reagent | pRRLSIN.cPPT.PGK-GFP.WPRE | Didier Trono Lab | Addgene#12252 | Constitutive lentiviral expression vector (empty backbone) |
| Recombinant DNA reagent | pCW57.1 | David Root Lab | Addgene#41393 | Doxycycline-inducible lentiviral expression vector (empty backbone) |
| Recombinant DNA reagent | pCW57.1-3xFLAG-CIC | This study | N/A | Lentiviral expression plasmid for doxycycline-inducible transgene expression |
| Recombinant DNA reagent | pCW57.1-3xFLAG-CIC/DUX4 | This study | N/A | Lentiviral expression plasmid for doxycycline-inducible transgene expression |
| Recombinant DNA reagent | pCW57.1-3XFLAG-NLS-NLS-Dux | This study | N/A | Lentiviral expression plasmid for doxycycline-inducible transgene expression |
| Recombinant DNA reagent | pCW57.1-3XFLAG-NLS-NLS-DUX4 | This study | N/A | Lentiviral expression plasmid for doxycycline-inducible transgene expression |
| Recombinant DNA reagent | pCW57.1-3XFLAG-NLS-NLS-DUX4-dL2 | This study | N/A | Lentiviral expression plasmid for doxycycline-inducible transgene expression |
| Recombinant DNA reagent | pCW57.1-3XFLAG-NLS-NLS-DUX4-F67A | This study | N/A | Lentiviral expression plasmid for doxycycline-inducible transgene expression |
| Recombinant DNA reagent | pCW57.1-3XFLAG-NLS-NLS-DUX4-mL1 | This study | N/A | Lentiviral expression plasmid for doxycycline-inducible transgene expression |
| Recombinant DNA reagent | pCW57.1-3XFLAG-NLS-NLS-DUX4-mL1dL2 | This study | N/A | Lentiviral expression plasmid for doxycycline-inducible transgene expression |
| Recombinant DNA reagent | pCW57.1-3XFLAG-NLS-NLS-DUX4(aa339-424) | This study | N/A | Lentiviral expression plasmid for doxycycline-inducible transgene expression |
| Recombinant DNA reagent | pCW57.1-3XFLAG-NLS-NLS-DUX4aa154-271 | This study | N/A | Lentiviral expression plasmid for doxycycline-inducible transgene expression |
| Recombinant DNA reagent | pCW57.1-3XFLAG-NLS-NLS-DUX4aa154-308 | This study | N/A | Lentiviral expression plasmid for doxycycline-inducible transgene expression |
| Recombinant DNA reagent | pCW57.1-3XFLAG-NLS-NLS-DUX4aa154-372 | This study | N/A | Lentiviral expression plasmid for doxycycline-inducible transgene expression |
| Recombinant DNA reagent | pCW57.1-3XFLAG-NLS-NLS-DUX4CTD | This study | N/A | Lentiviral expression plasmid for doxycycline-inducible transgene expression |

*Appendix 1 Continued on next page*

*Appendix 1 Continued*

| Reagent type (species) or resource | Designation | Source or reference | Identifiers | Additional information |
|---|---|---|---|---|
| Recombinant DNA reagent | pCW57.1-3XFLAG-NLS-NLS-DUX4CTDmL1dL2 | This study | N/A | Lentiviral expression plasmid for doxycycline-inducible transgene expression |
| Recombinant DNA reagent | pCW57.1-3XFLAG-NLS-NLS-DUXB | This study | N/A | Lentiviral expression plasmid for doxycycline-inducible transgene expression |
| Recombinant DNA reagent | pCW57.1-3XFLAG-NLS-NLS-DUXBCTD | This study | N/A | Lentiviral expression plasmid for doxycycline-inducible transgene expression |
| Recombinant DNA reagent | pCW57.1-3xMYC-STAT1 | This study | N/A | Lentiviral expression plasmid for doxycycline-inducible transgene expression |
| Recombinant DNA reagent | pCW57.1-3xMYC-STAT1-S727A | This study | N/A | Lentiviral expression plasmid for doxycycline-inducible transgene expression |
| Recombinant DNA reagent | pCW57.1-3xMYC-STAT1-Y701A | This study | N/A | Lentiviral expression plasmid for doxycycline-inducible transgene expression |
| Recombinant DNA reagent | pRRLSIN-3XFLAG-NLS-NLS-DUX4CTD | This study | N/A | Lentiviral expression plasmid for constitutive transgene expression |
| Recombinant DNA reagent | pRRLSIN-3XFLAG-NLS-NLS-DUXBCTD | This study | N/A | Lentiviral expression plasmid for constitutive transgene expression |
| Sequence-based reagent | IFIH1_F | *Geng et al., 2012*. Dev Cell. doi: 10.1016/j.devcel.2011.11.013. | RT-qPCR primers | CTAGCCTGTTCTGGGGAAGA |
| Sequence-based reagent | IFIH1_R | *Geng et al., 2012*. Dev Cell. doi: 10.1016/j.devcel.2011.11.013. | RT-qPCR primers | AGTCGGCACACTTCTTTTGC |
| Sequence-based reagent | ISG20_F | *Geng et al., 2012*. Dev Cell. doi: 10.1016/j.devcel.2011.11.013. | RT-qPCR primers | GAGCGCCTCCTACACAAGAG |
| Sequence-based reagent | ISG20_R | *Geng et al., 2012*. Dev Cell. doi: 10.1016/j.devcel.2011.11.013. | RT-qPCR primers | CGGATTCTCTGGGAGATTTG |
| Sequence-based reagent | h16q21_F | *Maston et al., 2012* | ChIP-qPCR primers (gene desert region) | AAACAAGCATCAGGGTGGAC |
| Sequence-based reagent | h16q21_R | *Maston et al., 2012* | ChIP-qPCR primers (gene desert region) | GATCCCACAAAGGAAAGGAAC |
| Sequence-based reagent | GBP1_F | Origene Cat# HP205803 | RT-qPCR primers | TAGCAGACTTCTGTTCCTACATCT |
| Sequence-based reagent | GBP1_R | Origene Cat# HP205803 | RT-qPCR primers | CCACTGCTGATGGCATTGACGT |
| Sequence-based reagent | CXCL10_F | Primer Bank ID 323422857c1, https://pga.mgh.harvard.edu/primerbank, *Wang et al., 2012*. Nucleic Acids Res. doi: 10.1093/nar/gkr1013. | RT-qPCR primers | GTGGCATTCAAGGAGTACCTC |
| Sequence-based reagent | CXCL10_R | Primer Bank ID 323422857c1, https://pga.mgh.harvard.edu/primerbank, *Wang et al., 2012*. Nucleic Acids Res. doi: 10.1093/nar/gkr1013. | RT-qPCR primers | TGATGGCCTTCGATTCTGGATT |
| Sequence-based reagent | IDO1_F | PrimerBank ID 323668304c1, https://pga.mgh.harvard.edu/cgi-bin/primerbank/new_search2.cgi, *Wang et al., 2012*. Nucleic Acids Res. doi: 10.1093/nar/gkr1013. | RT-qPCR primers | GCCAGCTTCGAGAAAGAGTTG |
| Sequence-based reagent | IDO1_R | PrimerBank ID 323668304c1, https://pga.mgh.harvard.edu/cgi-bin/primerbank/new_search2.cgi, *Wang et al., 2012*. Nucleic Acids Res. doi: 10.1093/nar/gkr1013. | RT-qPCR primers | ATCCCAGAACTAGACGTGCAA |
| Sequence-based reagent | CXCL10_F | *Rosowski et al., 2014* | ChIP-qPCR primers | AAAGGAACAGTCTGCCCTGA |
| Sequence-based reagent | CXCL10_R | *Rosowski et al., 2014* | ChIP-qPCR primers | GCCCTGCTCTCCCATACTTT |
| Sequence-based reagent | GBP1_F | *Rosowski et al., 2014* | ChIP-qPCR primers | TGGACAAATTCGTAGAAAGACTCA |

*Appendix 1 Continued*

| Reagent type (species) or resource | Designation | Source or reference | Identifiers | Additional information |
|---|---|---|---|---|
| Sequence-based reagent | GBP1_R | *Rosowski et al., 2014* | ChIP-qPCR primers | GCACAAAAACTGTCCCCAAC |
| Sequence-based reagent | IDO1_F | *Rosowski et al., 2014* | ChIP-qPCR primers | CACAGTCATTGTATTCTCTTTGCTG |
| Sequence-based reagent | IDO1_R | *Rosowski et al., 2014* | ChIP-qPCR primers | GCATATGGCTTTCGTTACAGTC |
| Sequence-based reagent | CD74_F | UCSC Genome Browser, *Zeisel et al., 2013*. Bioinformatics. doi: 10.1093/bioinformatics/btt145. | RT-qPCR primers | CGCGACCTTATCTCCAACAA |
| Sequence-based reagent | CD74_R | UCSC Genome Browser, *Zeisel et al., 2013*. Bioinformatics. doi: 10.1093/bioinformatics/btt145. | RT-qPCR primers | CAGGATGGAAAAGCCTGTGT |
| Sequence-based reagent | CXCL9_F | UCSC Genome Browser, *Zeisel et al., 2013*. Bioinformatics. doi: 10.1093/bioinformatics/btt145. | RT-qPCR primers | TCTTTTCCTCTTGGGCATCA |
| Sequence-based reagent | CXCL9_R | UCSC Genome Browser, *Zeisel et al., 2013*. Bioinformatics. doi: 10.1093/bioinformatics/btt145. | RT-qPCR primers | TAGTCCCTTGGTTGGTGCTG |
| Transfected construct (human) | Control (non-sil.) siRNA | QIAGEN | Cat# 1022076 | Non-targeting control siRNA |
| Transfected construct (human) | FlexiTube siRNA Hs_CIC_6 | QIAGEN | Cat# SI04275656 | siRNA targeting CIC |
| Transfected construct (human) | FlexiTube siRNA Hs_CIC_8 | QIAGEN | Cat# SI04368469 | siRNA targeting CIC |
| Transfected construct (human) | GeneSolution siRNA Hs_DUX4_11 | QIAGEN | Cat# SI04239753 | siRNA targeting DUX4. |

