## [Editor Report]

In this study, the authors provide convincing data to demonstrate that the transcription factor DUX4 functions as a negative regulator of interferon signaling by inhibiting STAT1, thereby suppressing interferon-stimulated gene induction. These studies are important in revealing a critical mechanistic link between DUX4 expression and the modulation of immune signaling pathways. As DUX4 is emerging as a key molecule in early mammalian development and in diverse pathologies including muscular dystrophy and solid tumors, this study will be of broad interest to the fields of development, cancer, and immunology.

---

## [Decision Letter]

**Decision letter after peer review:**

Thank you for submitting your article "Human DUX4 and mouse Dux interact with STAT1 and broadly inhibit interferon-stimulated gene induction" for consideration by *eLife*. Your article has been reviewed by 2 peer reviewers, and the evaluation has been overseen by a Reviewing Editor and Carla Rothlin as the Senior Editor. The following individual involved in the review of your submission has agreed to reveal their identity: Vittorio Sartorelli (Reviewer #1).

Essential revisions:

Both reviewers agreed that the manuscript would be strengthened by demonstrating the key findings with new experiments in a more biologically relevant system. Specifically,

1) Please plan to address Reviewer #1, Comment #2, and Reviewer #2, Comment #4 with additional experiments.

2) Please address all other reviewer points with clarification or discussion, as requested, or with additional experiments if appropriate. It would be helpful to include a section on limitations in the Discussion to address comments such as Reviewer #1, Comment #3.

*Reviewer #1 (Recommendations for the authors):*

1. Figure 4. It seems as if IFN-γ treatment increases the expression of the STAT1 and STAT1 S727A vectors (see Inputs). If my interpretation of the data is correct, the IP'ed STAT1 and STAT1 S727A should be corrected for their respective inputs.

2. Figure 6. Documenting the interaction of CIC-DUX4 with STAT1 in Kitra-SRS cells would reinforce the present findings.

3. The interesting finding that DUX4 prevents STAT1 binding and Pol-II recruitment may be mechanistically clarified by employing DUX4 and Gal4-Sp1 expression vectors with a Gal4-reporter followed by ChIP-qPCR with Gal4 and Pol-II antibodies.

*Reviewer #2 (Recommendations for the authors):*

1. Statistical analyses are missing in most panels, please add and also add "non-significant" where important to interpret the data (i.e. Figure 4C).

2. Please provide a rationale for using DUX4-CTD from Figure 3 (related to the concern of missing regulatory elements). There may be regulatory domains missing from the full-length protein.

3. The conclusion in line 242 that the STAT1-DUX4-CTD interaction is happening in the nucleus is not possible with the data provided. Making this claim will require additional experiments, i.e. pulldowns and /or colocalization studies with DUX4 lacking an NLS.

4. Validation of key findings of STAT1-DUX4 interaction with a system not relying on doxycycline, and using different cells, preferentially primary cells that do not at steady-state have total STAT1 in the nucleus, is needed.

5. Description of protein domains and mutants in Figure 2 is not clear. Please add a more detailed description of the different annotated domains and information on whether the NLS is the same as endogenous NLS and information on whether this NLS has been mapped functionally or by homology.

6. Figure 1B, naming "RIG-I" is misleading, as not RIG-I is provided, but a ligand for RIG-I. Suggest renaming to "3'ppp-dsRNA".

7. A lot of pressure lies on the data in Figure 6A, i.e. the physiological relevance statement made in line 387. While this is a beautiful experiment, the phenotype is very difficult to spot with the data provided and does not seem consistent across events marked with an arrow. Suggest quantification of the events with a higher number of cells and biological repeats to allow for statistical analysis and drive this conclusion home.

---

## [Author Response]

Reviewer #1 (Recommendations for the authors):1. Figure 4. It seems as if IFN-γ treatment increases the expression of the STAT1 and STAT1 S727A vectors (see Inputs). If my interpretation of the data is correct, the IP'ed STAT1 and STAT1 S727A should be corrected for their respective inputs.

We agree that the Input samples on the Western blot of Figure 4B probed with α-STAT1 might suggest a change in transgene expression with IFNγ treatment. However, the IFN-γ-treated samples from each transgenic cell line have comparable transgene expression as seen by both the α-STAT1 and α-Myc signals. Rather than computationally correcting for untreated-to-IFNγ expression levels in these samples, we have revised the text to acknowledge the reviewers point and direct the reader to compare the IFNγ-treated samples. We have added text edits to clarify this comparison in lines 202-206.

2. Figure 6. Documenting the interaction of CIC-DUX4 with STAT1 in Kitra-SRS cells would reinforce the present findings.

We have added additional data from the Kitra-SRS cell lines, including both a Proximity Ligation Assay (PLA) and a flow cytometry experiment.

In our PLA, we compared the PLA signal resulting from CIC-DUX4/STAT1 or CIC-DUX4/pSTAT1-Y701 interaction in untreated or IFNγ-treated Kitra-SRS cells (Figure 6C). We counted the number of PLA foci per nucleus in each condition and antibody pair and plotted these data. We found that there was a significant increase in the nuclear interaction of CIC-DUX4 and both total STAT1 and pSTAT1-Y701 with IFNγ treatment, consistent with our results suggesting that the CIC-DUX4 fusion protein is interacting with STAT1 in the nuclei of IFNγ-treated cells and interfering with its ability to upregulate ISGs.

In our flow cytometry experiment, we treated both MB135 parental cells (which endogenously express full-length CIC) and Kitra-SRS cells (expressing the CIC-DUX4 fusion protein) with either control siRNAs (siCTRL) or siRNAs targeting CIC and DUX4 (siCIC-DUX4) in the absence or presence of IFNγ (Figure 6—Figure Supplement 1). While knockdown of endogenous CIC in MB135 cells had no effect on MHC-I upregulation in IFNγ-treated cells, knockdown of the CIC-DUX4 fusion in Kitra-SRS cells increased MHC-I positive cells from 27.9% to 48.1%, suggesting that the CIC-DUX4 fusion protein interferes with MHC-I upregulation in this cell type.

Together, these provide additional data documenting the interaction of CIC-DUX4 with STAT1 in Kitra-SRS cells and further show that interaction suppresses MHC-I surface presentation. We thank the reviewer for suggesting adding such experiments to reinforce the original findings.

3. The interesting finding that DUX4 prevents STAT1 binding and Pol-II recruitment may be mechanistically clarified by employing DUX4 and Gal4-Sp1 expression vectors with a Gal4-reporter followed by ChIP-qPCR with Gal4 and Pol-II antibodies.

We appreciate the reviewer’s suggestion to use Gal4-Sp1 expression vectors to clarify the mechanism by which DUX4 prevents STAT1 binding and Pol-II recruitment. We have added a final paragraph to the Discussion to acknowledge that additional studies, such as the Gal4-Sp1 experiment suggested by this reviewer, will be necessary to clarify the mechanism(s) of preventing Pol-II recruitment.

Reviewer #2 (Recommendations for the authors):1. Statistical analyses are missing in most panels, please add and also add "non-significant" where important to interpret the data (i.e. Figure 4C).

We thank the reviewer for this suggestion and have accordingly added statistical analyses to all comparisons in Figure 1, Figure 2, Figure 4C, Figure 5A, Figure 6B, Figure 6C, Figure 7, and Figure 1—Figure Supplement 2.

2. Please provide a rationale for using DUX4-CTD from Figure 3 (related to the concern of missing regulatory elements). There may be regulatory domains missing from the full-length protein.

We appreciate the reviewer’s concern. Figures 1 and 2 used both full-length DUX4 constructs and different mutants and deletions to map the necessary and sufficient domains for inhibition of ISG induction. For the discovery of interacting proteins, we decided to use the DUX4-CTD because it contained both the necessary and sufficient region. While we agree there might be other regulatory domains missing, we wanted to first identify the factors interacting with the necessary and sufficient regions. We hope future studies can identify the roles of other domains and factors. We have added additional text to clarify our rationale in lines 170-172.

3. The conclusion in line 242 that the STAT1-DUX4-CTD interaction is happening in the nucleus is not possible with the data provided. Making this claim will require additional experiments, i.e. pulldowns and /or colocalization studies with DUX4 lacking an NLS.

We understand the potential for confusion and thank the reviewer for the opportunity to clarify our point in the text. While we are not arguing against the formal possibility of interactions in the cytoplasm, we are arguing for the presence and importance of interactions between the DUX4-CTD and STAT1 in the nucleus. We have made edits to lines 214-218, accordingly.

4. Validation of key findings of STAT1-DUX4 interaction with a system not relying on doxycycline, and using different cells, preferentially primary cells that do not at steady-state have total STAT1 in the nucleus, is needed.

We agree with the reviewer that it is important to show that treatment with doxycycline is not contributing to the suppression of ISGs. In our original submission, and also retained in our revised manuscript, we show in Figure 1 that doxycycline induction of iDUX4 suppresses ISGs, whereas doxycycline treatment of parental MB135 cells or MB135-iDUXB cells does not suppress ISGs. And in Figure 2 we show that doxycycline induction of the iDUX4 (L)LxxL(L) mutant (iDUX4-CTDmL1dL2 / iDUX4-mL1dL2) does not suppress ISG induction. Therefore, we interpret the reviewer’s concern as focused on the nuclear/cytoplasmic distribution of STAT1 in the immortalized cells and demonstrating the interaction of STAT1 and DUX4-CTD in other cells and/or primary cells.

Together with data from the original submission, we have also added data to the revised manuscript to show that the distribution of STAT1 in the immortalized MB135iDUX4 cells is similar to that in primary cells. As shown in the original submission, Figure 4—Figure Supplement 1A and B show low level cytoplasmic and nuclear distribution of STAT1 in the absence of IFNγ and enhanced overall STAT1 expression and nuclear accumulation in the presence of IFNγ. In the revision, we have added a Figure 4—Figure Supplement 1C to show that pSTAT1(Y701) is only detectable with IFNγ treatment and is nuclear; and we have added Figure 4—Figure Supplement 1D showing the distribution of total STAT1 in primary human fibroblasts, primary MB135 myoblasts, and immortalized MB135 myoblasts is similar to the distribution in the immortalized MB135iDUX4 myoblasts, which is also the same total STAT1 distribution described for different cell types in the Human Protein Atlas (http://www.proteinatlas.org). Together, these data show that the distribution of STAT1 in the MB135iDUX4 cells is not abnormal but is similar to that in primary cells and other cell types.

To validate the DUX4 STAT1 interaction in a primary cell system not relying on doxycycline, we transduced primary human fibroblasts with lentiviral constructs constitutively expressing either 3XFLAG-DUX4-CTD or 3XFLAG-DUXB-CTD and performed a Proximity Ligation Assay (PLA) with anti-FLAG and anti-STAT1 antibodies, as we had done to show the interaction of STAT1 with the DUX4-CTD in the MB135iDUX4 cells in the original submission (see Figure 4C). Figure 4—Figure Supplement 2 shows a significantly higher PLA signal in nuclei expressing the DUX4-CTD compared to no transgene or the DUXB-CTD transgene and validates the DUX4-STAT1 interaction in primary cells in the absence of doxycycline.

5. Description of protein domains and mutants in Figure 2 is not clear. Please add a more detailed description of the different annotated domains and information on whether the NLS is the same as endogenous NLS and information on whether this NLS has been mapped functionally or by homology.

We thank the reviewer for bringing this to our attention. We have described the protein domains, mutations, and NLS more clearly in the legend of Figure 1—Figure Supplement 1 and referred to this description in the legend of Figure 2. The DUX4 nls domains have not been well characterized and map, in part, to the homeodomains. Therefore we used two NLS domains: the SV40 nls and the eight amino-acid nls from the SMCHD1 protein, as now described in the Figure1—Figure Supplement 1.

6. Figure 1B, naming "RIG-I" is misleading, as not RIG-I is provided, but a ligand for RIG-I. Suggest renaming to "3'ppp-dsRNA".

We thank the reviewer for pointing this out. We have edited both Figure 1B and the corresponding figure legend to reflect that our experiments were using the RIG-I ligand 5’ppp-dsRNA, not RIG-I itself.

7. A lot of pressure lies on the data in Figure 6A, i.e. the physiological relevance statement made in line 387. While this is a beautiful experiment, the phenotype is very difficult to spot with the data provided and does not seem consistent across events marked with an arrow. Suggest quantification of the events with a higher number of cells and biological repeats to allow for statistical analysis and drive this conclusion home.

We agree that some of the pressure of proving physiological relevance lies with Figure 6A (though we note that the Kitra-SRS cells, which have endogenous expression of a CIC-DUX4 fusion protein, also carry much of the weight of physiological relevance). We have moved these data to Figure 7A to make space for additional Kitra-SRS data in Figure 6, and we have quantified the microscopy images appropriately using Mean Fluorescent Intensity (MFI) analysis to show the mutual-exclusivity of IDO1 and DUX4 signal. Through our MFI analysis, we show quantitatively that for all nuclei imaged DUX4-signal is significantly lower in nuclei expressing IDO1 (IDO1+), and IDO1-signal is significantly lower in nuclei of myotubes expressing DUX4 (DUX4+). This supports our earlier statement that DUX4-expressing myotubes do not activate ISGs, such as IDO1, as robustly as DUX4-negative myotubes.